# Origin and evolution of the nuclear auxin response system

Sumanth K Mutte[1†], Hirotaka Kato[1†], Carl Rothfels[2], Michael Melkonian[3], Gane Ka-Shu Wong[4,5,6], Dolf Weijers[1*]

[1]Laboratory of Biochemistry, Wageningen University, Wageningen, Netherlands; [2]Department of Integrative Biology, University of California, Berkeley, United States; [3]Botanical Institute, Cologne Biocenter, University of Cologne, Cologne, Germany; [4]Department of Biological Sciences, University of Alberta, Edmonton, Canada; [5]Department of Medicine, University of Alberta, Edmonton, Canada; [6]BGI-Shenzhen, Shenzhen, China

**Abstract** The small signaling molecule auxin controls numerous developmental processes in land plants, acting mostly by regulating gene expression. Auxin response proteins are represented by large families of diverse functions, but neither their origin nor their evolution is understood. Here, we use a deep phylogenomics approach to reconstruct both the origin and the evolutionary trajectory of all nuclear auxin response protein families. We found that, while all subdomains are ancient, a complete auxin response mechanism is limited to land plants. Functional phylogenomics predicts defined steps in the evolution of response system properties, and comparative transcriptomics across six ancient lineages revealed how these innovations shaped a sophisticated response mechanism. Genetic analysis in a basal land plant revealed unexpected contributions of ancient non-canonical proteins in auxin response as well as auxin-unrelated function of core transcription factors. Our study provides a functional evolutionary framework for understanding diverse functions of the auxin signal.

DOI: https://doi.org/10.7554/eLife.33399.001

*For correspondence:
dolf.weijers@wur.nl

[†]These authors contributed equally to this work

Competing interests: The authors declare that no competing interests exist.

## Introduction

Auxins are a group of structurally related chemical compounds that control a multitude of growth and developmental processes in plants. The most common, naturally occurring auxin is indole-3-acetic acid (IAA), but synthetic analogs such as 2,4-dichlorophenoxy acetic acid (2,4-D) have largely overlapping biological activities (*Woodward and Bartel, 2005*). While auxins have been shown to trigger rapid cellular events such as membrane hyperpolarization (*Bates and Goldsmith, 1983*; *Etherton, 1970*), calcium influx (*Monshausen et al., 2011*; *Schenck et al., 2010*), and changes in endocytosis (*Paciorek et al., 2005*; *Robert et al., 2010*), its activity in controlling growth and development appear to be mainly mediated by changes in gene expression via a nuclear auxin pathway (NAP). Perturbation of this gene regulatory pathway interferes with most, if not all, developmental responses (*Weijers and Wagner, 2016*). Indeed, in the moss *Physcomitrella patens*, it was shown that a complete knock-out mutant of this pathway does not show any transcriptional response to auxin (*Lavy et al., 2016*). The NAP encompasses three dedicated protein families (*Figure 1A,B*). Various auxins, including IAA and 2,4-D, are perceived by a co-receptor complex consisting of TRANSPORT INHIBITOR RESPONSE 1/AUXIN SIGNALING F-BOX (TIR1/AFB) and AUXIN/INDOLE-3-ACETIC ACID (Aux/IAA) proteins (*Dharmasiri et al., 2005*; *Kepinski and Leyser, 2005*; *Tan et al., 2007*). Subsequent ubiquitination of the Aux/IAA proteins causes their degradation in the 26S proteasome (*Gray et al., 2001*). When not degraded, Aux/IAA proteins bind to and inhibit DNA-binding transcription factors, the AUXIN RESPONSE FACTORS (ARF) (*Kim et al., 1997*). Thus, auxin de-

**eLife digest** Across all kingdoms of life, signaling molecules like hormones, for example, control many aspects of the lives of organisms, including how they grow and develop. Cells have dedicated proteins that can recognize the signaling molecules, relay the information, and respond to the signal, for example by switching genes on or off. Such response systems usually consist of multiple components, and, throughout evolution, these response components have regularly been copied such that many species have multiple different versions of each one.

Auxin is a plant hormone that controls virtually all growth and developmental processes in plants, including many yield traits in crops. However, no one knows why it is involved in so many processes. This is partly because it is not clear how the response system for this central signaling molecule was first born, or how it has increased in its complexity.

To address this, Mutte, Kato et al. explored the genetic information of more than a thousand plant species, including algae, which span more than 700 million years of evolution. Their analysis showed that all auxin response components were assembled from pieces of much older genes, but that they first came together when plants conquered land. Indeed, the auxin response appears to have developed on top of a pre-existing genetic regulator that is still present in modern-day algae. Mutte, Kato et al. then used experiments to show how stepwise increases in the number and types of auxin response components have shaped sophisticated, complex responses in land plants, and to demonstrate how ancient components control auxin response.

Together these findings provide a framework for understanding the many functions of auxin in plants, and how this came to be. They also show how complexity can be accomplished in a signal response pathway, and how diversity evolves in gene families. Similar studies on other response systems in plants and beyond are likely to help reveal common principles of hormone response evolution and diversification of gene regulation systems.

DOI: https://doi.org/10.7554/eLife.33399.002

represses ARFs, allowing these to activate or repress their direct target genes (*Ulmasov et al., 1999*).

A central question in plant biology is how this simple transcriptional system with only three dedicated components can generate a multitude of local auxin responses to support various developmental functions. In flowering plants such as *Arabidopsis thaliana*, it is likely that the size of TIR1/AFB (six members), Aux/IAA (29 members) and ARF (23 members) gene families allows combinatorial assembly of distinct, local auxin response pathways. Given that diversity in auxin responses follows from diversification in its response proteins, it is still unclear how NAP complexity evolved from simpler ancestral states. Furthermore, while intuitive, a key question is whether increased NAP complexity indeed enabled more complex and diverse auxin responses during plant evolution. A third important question is where, when, and from what precursors the NAP originated.

Eukaryotic photosynthetic organisms diverged into three groups, Glaucophyta, Rhodophyta (red algae), and Viridiplantae more than 1.5 billion years ago (*Yoon et al., 2004*). Viridiplantae are further classified into chlorophyte algae and streptophytes, which include charophyte algae and land plants. Bryophytes represent the earliest diverging land plants and consist of three groups: hornworts, liverworts and mosses. After the split from bryophytes, ancestral vascular plants changed their life cycle from haploid-dominant to diploid-dominant and established a vascular system and root architecture, forming the group of lycophytes and euphyllophytes (ferns, gymnosperms and angiosperms).

The presence of a functional NAP with reduced genetic redundancy has been reported in model bryophytes (*Flores-Sandoval et al., 2015*; *Kato et al., 2015*; *Prigge et al., 2010*; *Rensing et al., 2008*), whereas the presence of endogenous auxin is also reported in wide range of algal species (*Žižková et al., 2017*). Thus, a prediction is that the auxin response system may predate land plants, and that complexity evolved after the divergence of ancestral vascular plants from bryophytes. A key challenge is to identify the origin of the NAP system, as well as to reconstruct the steps in the evolution of its complexity. However, only little genome data are currently available from non-flowering land plants (*Rensing, 2017*), which makes such inferences extremely challenging. In addition, studies using only selected model species bear the risk of generalizing observations from non-representative

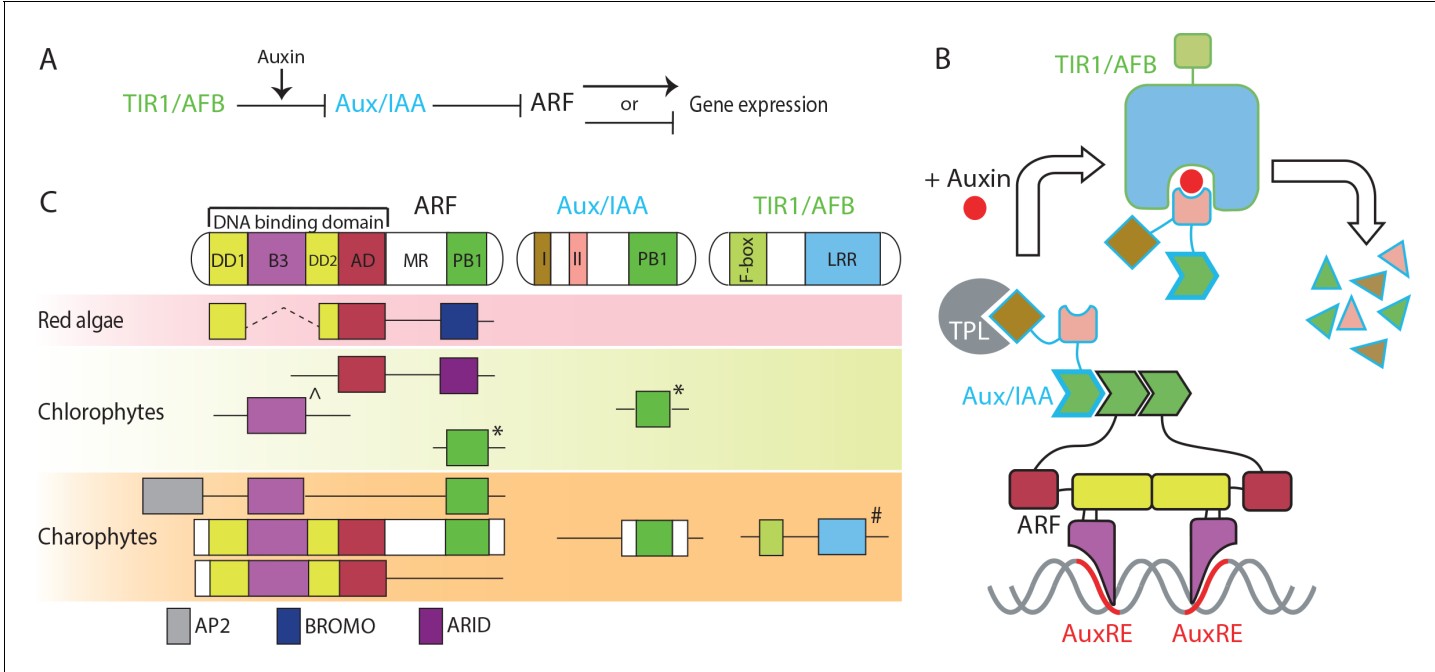

**Figure 1.** Proteins in nuclear auxin pathway; mechanism and origin of the domains. (A, B) Scheme of NAP in land plants. In the absence of auxin, Aux/IAA inhibit ARF via their PB1 domains, and by recruiting the TPL co-repressor. Auxin stabilizes the interaction between Aux/IAA and TIR1/AFB, followed by proteasome-mediated degradation of Aux/IAA. (C) Domain structure of NAP components in land plants and presence of each domain in algae, as recovered in transcriptomes. (Sub-)domains are indicated by colors, that match those in (B). ̂: basal to all B3-type transcription factors in land plants, *: difficult to assign to ARF or Aux/IAA family; #: forming basal clade to both TIR1/AFB and COI1 in land plants.

DOI: https://doi.org/10.7554/eLife.33399.003

The following figure supplements are available for figure 1:

**Figure supplement 1.** The work flow of phylogenetic tree construction.

DOI: https://doi.org/10.7554/eLife.33399.004

**Figure supplement 2.** Phylogenetic tree of ARF and RAV proteins.

DOI: https://doi.org/10.7554/eLife.33399.005

**Figure supplement 3.** DNA-binding domain of RAV proteins.

DOI: https://doi.org/10.7554/eLife.33399.006

genomes, due to species-specific gene-duplication, -loss, and -diversification. Therefore, it is necessary to analyze multiple species to understand evolutionary trends.

Here, we describe a deep phylogenomic analysis of NAP components using a large transcriptome dataset with more than 1000 plant species including many algae. This extensive dataset enabled us to reconstruct the ancestral states of auxin response gene families at key nodes in plant evolution. We infer plausible origins and evolutionary patterns for each auxin response gene family and predict auxin response properties at evolutionary nodes. Using comparative RNA-seq of six species, we tested and extended these predictions. Finally, we used a genetic strategy in a bryophyte to demonstrate surprising non-contributions of an ancient ARF class as well as contribution of deeply conserved non-canonical NAP components to auxin signaling. Our work provides a deep view into early steps in the origin, evolution and design principles of the multi-functional auxin response system.

## Results

### A phylogenomic strategy for reconstructing ancestral states

To reconstruct origin and early diversification in auxin response gene families, we designed a strategy (*Figure 1—figure supplement 1*) that uses a large transcriptome dataset (OneKP) including multiple species for each major branch in plant species phylogeny (*Matasci et al., 2014*). The depth and quality of each individual RNA-seq-derived transcriptome is limited, and a further caveat of

transcriptome-based gene identifications is that the number of genes may be underestimated if a gene is not expressed under the sampling conditions or in the sampled tissue. However, the availability of transcriptomes from multiple tissue samples of multiple related species, should allow deduction of the ancestral state that defines the gene complement at each evolutionary node. It should be stressed that this number represents the ancestral state at a given node, and species-specific gene duplications and gene losses will have modified the gene complement in individual species. Given our focus on early events in nuclear auxin response evolution, we have used all available transcriptomes of red algae, green algae, bryophytes, lycophytes, ferns, and gymnosperms from the OneKP dataset (*Supplementary file 1*). We also included all available angiosperm species in the Chloranthales, Magnoliids and ANA grade, as well as several species in both monocots and dicots (*Supplementary file 1*). For reference and quality control purposes, we included genome-based sequences from well annotated model species.

## Origin of nuclear auxin response components

Each of the three auxin response protein types (ARFs, Aux/IAAs, and TIR1/AFBs) are multi-domain proteins and we initially focused on the origin of these proteins. Therefore, we asked where domains, or parts thereof, were found, and at what node the multi-domain proteins first appear.

ARF proteins carry an N-terminal DNA-binding domain (DBD) which consists of a composite dimerization domain (DD; made up of two separate subdomains [DD1 and DD2] that fold into a single unit), a B3-type DNA-interaction domain, and an ancillary domain (AD) of unknown function (*Figure 1C*; *Boer et al., 2014*). In land plants, the DD and AD are only found in the ARF family. The C-terminal Phox and Bem 1 (PB1) domain is shared among ARF and Aux/IAA proteins and mediates homo- and hetero-oligomerization (*Korasick et al., 2014*; *Nanao et al., 2014*). Finally, ARFs contain a less well-defined Middle Region (MR) separating the PB1 and DBD (*Figure 1C*). In red algae, we found proteins containing an N-terminal portion of DD1, DD2, and AD, lacking a B3 or PB1 domain, but instead flanked by a C-terminal bromodomain (BROMO; InterPro ID: IPR001487; *Figure 1C*). The DD1 and DD2 motifs in red algae are spaced by 20–30 conserved amino acids, which is much shorter than the B3 domain (~120 amino acids; *Supplementary file 2*). In chlorophytes, we found a protein with only AD, flanked by a DNA-binding AT-rich interaction domain (ARID; InterPro ID: IPR001606; *Figure 1C*). Furthermore, we found separate proteins that either represented a B3 or a PB1 domain (*Figure 1C*). Thus, all ARF subdomains had been established before the split of the streptophytes, but not combined in a single protein. In contrast, we discovered full-length ARF-like proteins containing a DBD with a B3 domain inserted between DD and AD in charophytes (*Figure 1C* and *Figure 1—figure supplement 2*). Land plant ARFs can be grouped into three classes, A, B and C (*Finet et al., 2013*). Based on transactivation assays, class A and B ARFs are classified as transcriptional activators and repressors, respectively (*Kato et al., 2015*; *Ulmasov et al., 1999*). Class C-ARFs are generally recognized as transcriptional repressors based on the amino acid composition of MR, but this has not yet been fully supported by experimental evidence (*Kato et al., 2018*). Phylogenetic analysis revealed that the ARF-like proteins in charophytes fall in two sister clades and likely represent separate precursors of class C-ARFs (proto-C-ARFs) and A/B-ARFs (proto-A/B-ARFs) of land plants (*Figure 2* and *Figure 1—figure supplement 2*). Interestingly, we found the PB1 domain only in proto-C-ARFs, which could, however, be due to sparse sampling in some charophyte lineages (*Figure 1—figure supplement 2*).

To understand if the proto-ARFs share conserved, functionally important residues, we generated homology models based on available DBD crystal structures of *A. thaliana* ARF1 and ARF5 (*Boer et al., 2014*). As no class C-ARF structure is known, we first modeled the *A. thaliana* ARF10 DBD to compare with proto-C-ARFs. Next, homology models for proto-ARFs in *Spirogyra pratensis* (SpARF; proto-C-ARF) and *Mesotaenium caldariorum* (McARF; proto-A/B-ARF) were generated. We also included all three ARFs of the bryophyte *M. polymorpha* (MpARF1-3) representing each major class, and compared all models to *A. thaliana* ARF structures. This analysis revealed that all proto-ARFs likely share a conserved structural topology (*Figure 3A*). Strikingly, all DNA-binding residues follow the spatial restraints needed for DNA binding in all ARFs tested, suggesting a conserved mode of DNA binding. On the other hand, dimerization residues are conserved only in the (proto-) A/B-ARFs (McARF, MpARF1, and MpARF2) but not in the (proto-)C-ARFs (SpARF, MpARF3, and ARF10). These results clearly demonstrate that canonical ARF proteins were established and differentiated into two classes in charophyte algae.

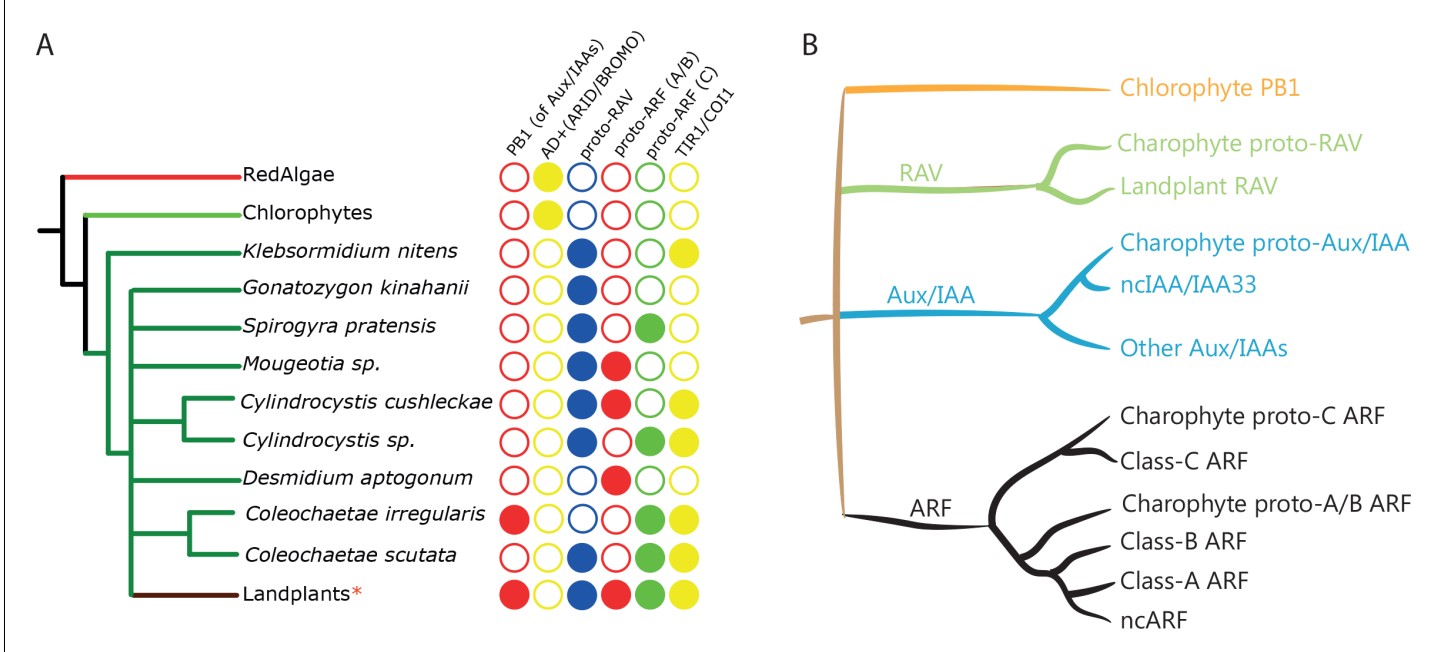

**Figure 2.** Distribution of auxin signaling proteins precursors in algal lineages. (A) Occurrence of NAP components in red algae, chlorophytes, and charophytes. Empty circles and filled circles indicate the absence and presence of that particular component, respectively. *: Land plants have defined three classes of ARFs, RAV without PB1, and separate TIR1/AFB and COI1 receptors. (B) Schematic illustration of the phylogenetic arrangement of RAV1, Aux/IAA and ARFs based on the DBD tree and PB1 tree. Note that only branches with strong bootstrap support are shown.

DOI: https://doi.org/10.7554/eLife.33399.007

The following figure supplement is available for figure 2:

**Figure supplement 1.** Phylogenetic tree based on PB1 domain.

DOI: https://doi.org/10.7554/eLife.33399.008

In addition to the proteins with canonical ARF-like structure, we found a group of charophyte proteins consisting of an AP2 DNA-binding domain along with B3 and PB1 domains (*Figure 1C*). Land plants also have a protein family containing AP2 domain in their N-terminus, followed by a B3 domain. These proteins are called REALATED TO ABI3 AND VP1 (RAV). Interestingly, land plant RAV proteins do not have a PB1 domain and it is known that the B3 domain of RAV and ARF binds different DNA sequences (*Boer et al., 2014*; *Matías-Hernández et al., 2014*). The B3 domain of RAV-like proteins in charophytes is much more similar to RAV's than to ARF's in land plants and phylogenetic analysis showed that the RAV-like proteins of charophytes position along with RAV family in land plants (*Figure 2B* and *Figure 1—figure supplements 2* and *3*). Thus, we classify these proteins as proto-RAV. In the charophyte green algae, the two classes of proto-ARFs and proto-RAVs are found in various combinations in each species (*Figure 2A*). While sequencing depth may be insufficient to detect all proto-ARFs and proto-RAVs, there does not appear to be a conserved pattern in the order of appearance and retention of these genes.

We next considered the origin of the Aux/IAA proteins. These proteins contain two functional small domains in addition to a C-terminal PB1 domain (*Figure 1B,C*). The N-terminal domain I recruits the TOPLESS (TPL) transcriptional co-repressor (*Szemenyei et al., 2008*). Domain II mediates the auxin-dependent interaction with TIR1/AFB and thus acts as a degron (*Dharmasiri et al., 2005*; *Gray et al., 2001*; *Kepinski and Leyser, 2005*). Because domain I and II are too small for reliable BLAST searches, we used the PB1 domain to identify potential family members. No PB1-containing proteins were identified in red algae, while we found proteins with a PB1 domain but no DBD in chlorophytes (*Figure 1C*). Phylogenetic analysis based on the PB1 domain indicated these are neither closely related to RAV, nor to Aux/IAA and ARF families (*Figure 2B* and *Figure 2—figure supplement 1*). PB1 domain-containing proteins that lack a DBD were also found in many of the charophyte algae (*Figures 1C* and *3B* and *Figure 2—figure supplement 1*). Most of them were

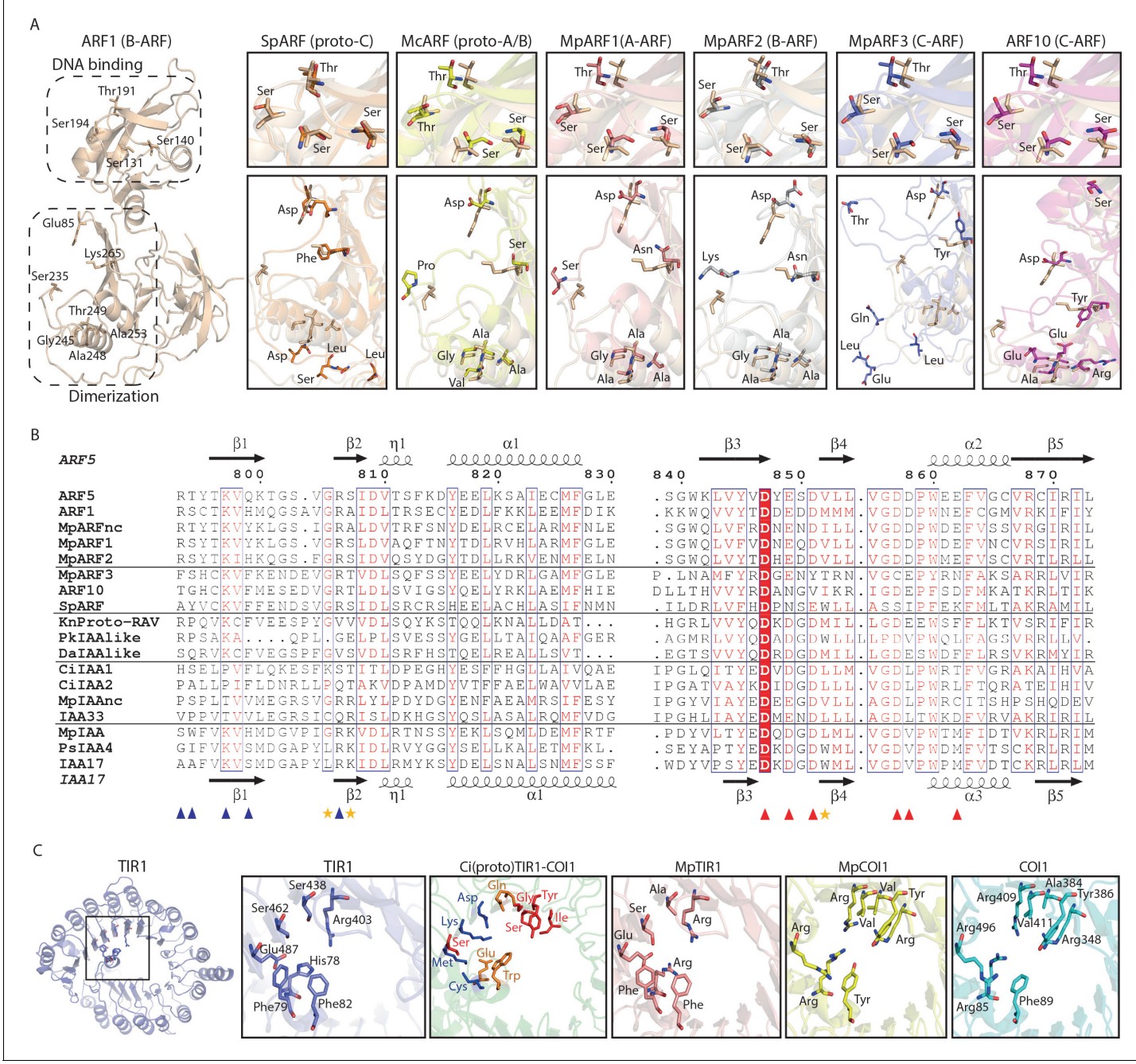

**Figure 3.** Homology models of ancestral ARF, Aux/IAA and TIR1/AFB proteins. (**A**) Homology models for ARF DBDs. The crystal structure of *Arabidopsis thaliana* ARF1-DBD is shown on the left with important residues for DNA binding (top) and dimerization (bottom). Homology models for (proto-)ARFs are overlaid on *A. thaliana* ARF1 in right panels (brown). (**B**) Alignment of PB1 domain of (proto-)ARF, Aux/IAA and proto-RAV proteins. Numbering is based on the ARF5 protein of *A. thaliana*. Arrows and helices indicate β-sheets and α-helices in ARF5 and IAA17 of *A.thaliana*, respectively. Blue and red triangles indicate positive (+) and negative (-) faces, respectively. Golden asterisks represent the residues of polar interactions. (**C**) Homology models for TIR1/AFB and COI1 proteins. Left panel shows crystal structure of *A. thaliana* TIR1 from top view. Auxin-binding pocket of TIR1/AFB and jasmonate-binding pocket of COI1 are shown in right panels. Hormone-binding residues are indicated as stick model in TIR1 and COI1 of land plants. Blue, red or orange residues in the model for the *Coleochaete irregularis* protein indicate the residues aligned with hormone binding residues of TIR1, COI1 or both, respectively. Ci: *Coleochaete irregularis*, Da: *Desmidium aptogonum*, Kn: *Klebsormidium nitens*, Mc: *Mesotaenium caldariorum*, Mp: *Marchantia polymorpha*, Pk: *Parachlorella kessleri*, Ps: *Pisum sativum*, Sp: *Spirogyra pratensis*.
DOI: https://doi.org/10.7554/eLife.33399.009

placed along with proto-RAV in phylogenetic tree, but the sequences from *Coleochaetae irregularis* were placed along with the Aux/IAA in land plants that is separate from the PB1 of both ARFs and proto-RAV proteins (*Figures 2B* and *3B* and *Figure 2—figure supplement 1*). Even though the N-terminal part of the PB1 domain is not as conserved as the C-terminal part, several critical residues were found to be conserved in Aux/IAA-like sequences (*Figure 3B*). These results indicate that the PB1 domain of land plant ARFs and Aux/IAAs had separate precursors in charophytes. We could, however, not detect domain I or II in Aux/IAA-like genes of charophyte algae, even when scrutinizing individual sequences. We thus conclude that Aux/IAA proteins with all three functional domains are limited to land plants.

Finally, we explored the origin of the TIR1/AFB auxin co-receptor that consist of an N-terminal F-box domain that anchors the protein to the other subunits in the SCF E3 ubiquitin ligase complex, and a C-terminal leucine-rich repeat (LRR) domain that contains the auxin-binding pocket. Auxin acts as a molecular glue to stabilize the interaction between TIR1/AFBs and Aux/IAAs (*Tan et al., 2007*). The closest homolog of the TIR1/AFB proteins in *A. thaliana* is CORONATINE INSENSITIVE 1 (COI1), which functions as a receptor of the jasmonic acid (JA) phytohormone (*Katsir et al., 2008*). In our homology search, we could not identify any proteins showing homology to either TIR1/AFB or COI1 in red algae and chlorophytes (*Figures 1C* and *2A*). We did find many proteins showing homology to TIR1/AFB and COI1 in the transcriptomes of charophyte algae (*Figures 1C* and *2A*). Phylogenetic analysis indicated that some of these proteins form a sister group to both TIR1/AFB and COI1 in land plants and none of the charophyte proteins are specifically grouped into either TIR1/AFB or COI1 clades (*Figure 4* and *Figure 4—figure supplement 2*), suggesting that charophytes had an ancestor that gave rise to both auxin and JA receptors. To infer whether the TIR1/AFB/COI1-like proteins of charophytes function as receptors for auxin or JA, we generated homology models of the TIR1/AFB/COI1-like protein from *C. irregularis* and the bryophyte *M. polymorpha* MpTIR1 and MpCOI1, using the *A. thaliana* TIR1 and COI1 crystal structures (*Sheard et al., 2010*; *Tan et al., 2007*) as templates for modeling. Even though the secondary structure of the *C. irregularis* protein was highly similar to that of land plant TIR1 and COI1 (*Supplementary file 2*), at the level of amino acid sequence, the protein did not resemble either TIR1/AFB or COI1. Out of 40 residues conserved in either TIR1/AFB's or COI1's, only 7 and 11 residues are identical to TIR1/AFBs and COI1s, respectively (*Supplementary file 2*; black stars). Notably, most of the hormone-contacting residues (11 out of 12) are different from both TIR1/AFB and COI1 (*Figure 3C* and *Supplementary file 2*). These results suggest that the charophyte TIR1/AFB/COI1 precursor may not act as an auxin or JA receptor, and we conclude that dedicated receptors for auxin and JA were established only in land plants. Taken together, our analyses suggest that the components of NAP were established in the common ancestor of land plants by combining pre-existing components and that the system evolved to regulate pre-existing transcription factors.

## Evolution of complexity in the nuclear auxin response system

All three gene families have evolved to considerable size and diversity in angiosperms, and this diversity is thought to underlie multifunctionality of auxin as a hormone. We next aimed to reconstruct the evolutionary history of auxin response components across all land plant lineages.

Consistent with previous descriptions (*Finet et al., 2013*), our phylogenetic analysis showed that all land plant ARFs are divided into three phylogenetic lineages (*Figure 4* and *Figure 1—figure supplement 2*). Within the class C lineage, we did not find any duplications in the ancestors of non-angiosperm species. The split that generated *A. thaliana* ARF10/16 and ARF17 likely occurred early in angiosperm evolution, while the PB1 domain was lost in the ARF17 group (*Figure 4* and *Figure 1—figure supplement 2*). The class A-ARF is represented by a single copy in bryophytes and lycophytes. We found that a subset of genes lacking the DBD diverged from class A-ARFs in early land plants, is missing in hornworts and has been retained in liverworts, mosses and lycophytes (non-canonical ARF, ncARF; *Figures 3B* and *4* and *Figure 2—figure supplement 1*). A further gene duplication event in the ancestor of euphyllophytes gave rise to two class A sub-families corresponding to *A. thaliana* ARF5/7/19 and ARF6/8, respectively. In the ancestor of seed plants a gene duplication caused differentiation between the *A. thaliana* ARF5 and ARF7/19 subfamilies (*Figure 4* and *Figure 1—figure supplement 2*). Finally, two gene duplication events in the ancestral angiosperms led to ARF6 and ARF8 and to a paralogue of ARF7/19, which was lost in *A. thaliana* (*Figure 4* and *Figure 1—figure supplement 2*).

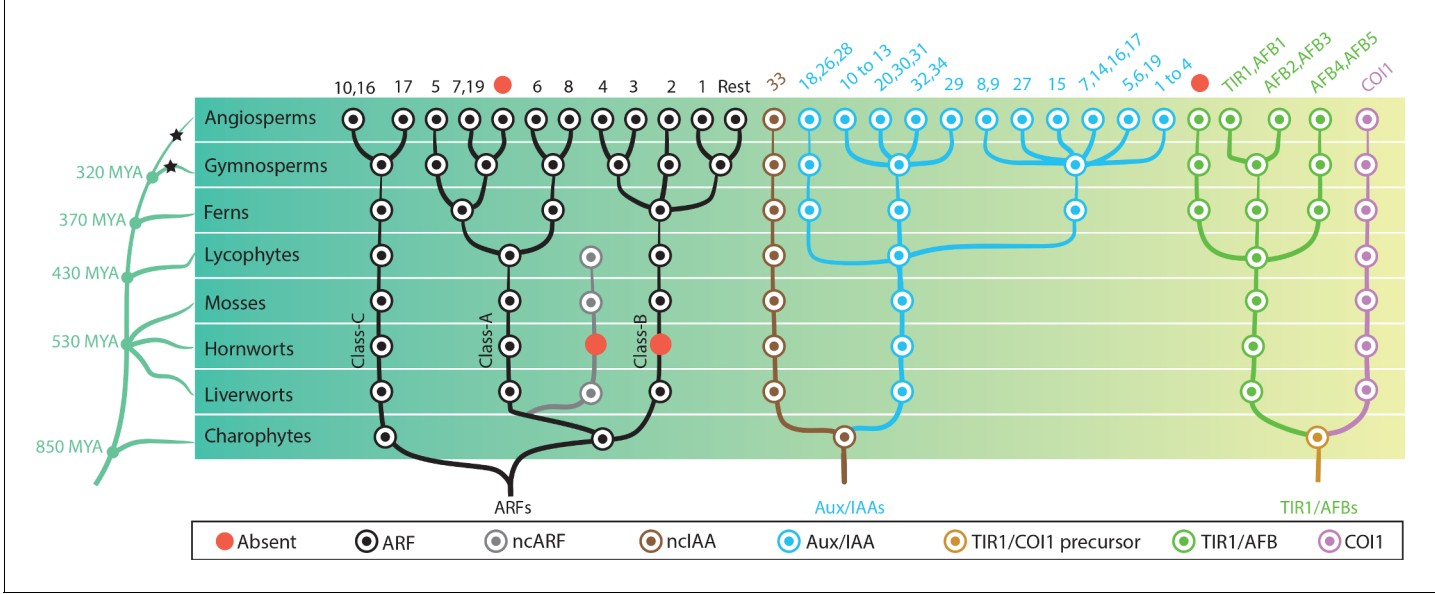

**Figure 4.** Reconstruction of ancestral state of NAP components in plant evolution. Phylogeny of taxonomic classes are shown in left. Time point of the lineage diversification was calculated using TimeTree database (*Kumar et al., 2017*). Black stars indicate whole genome duplication events (*Jiao et al., 2011*). Right: phylogenetic trees show the copy number and phylogenetic relationship of each protein family in the common ancestors. Each circle is colored according to protein type as indicated in the box. In the top row, numbers indicate which genes of *Arabidopsis thaliana* belong to each subfamily and red circles indicates missing subfamilies in *A. thaliana*. Note that only branches with strong bootstrap support are shown.

DOI: https://doi.org/10.7554/eLife.33399.010

The following figure supplements are available for figure 4:

**Figure supplement 1.** Phylogenetic tree of Aux/IAA.

DOI: https://doi.org/10.7554/eLife.33399.011

**Figure supplement 2.** Phylogenetic tree of the proteins containing F-box and LRR.

DOI: https://doi.org/10.7554/eLife.33399.012

**Figure supplement 3.** Phylogenetic tree of TIR1/AFB.

DOI: https://doi.org/10.7554/eLife.33399.013

Class B-ARFs are represented by a single gene in the ancestor of liverworts, mosses, lycophytes, and ferns. However, no hornwort species appears to contain class B-ARFs (*Figure 4* and *Figure 1—figure supplement 2*). Gene duplications in the ancestral gymnosperms gave rise to three class B-ARF copies, one representing *A. thaliana* ARF3/4, another leading to *A. thaliana* ARF2 and the third generating the remainder of the class B-ARFs in *A. thaliana* (*Figure 4* and *Figure 1—figure supplement 2*). Notably, the reported lack of the PB1 domain in ARF3 (*Finet et al., 2013*) is an independent loss in the common ancestor of monocots and eudicots (*Figure 1—figure supplement 2*).

Our data indicated that an ancestral Aux/IAA gene lacking domain I and II had been established during the evolution of charophytes, while 'true' Aux/IAAs with all functional domains are found only in land plants (*Figure 1C*). In addition to one copy of 'true' Aux/IAA, we found another set of deeply conserved non-canonical Aux/IAA-like sequences that lack the domain I and II (non-canonical Aux/IAA, ncIAA; *Figures 2B*, *3B* and *4*, *Figure 2—figure supplement 1*, and *Figure 4—figure supplement 1*). Strikingly, while the Aux/IAAs have diversified through gene duplications, the ncIAA is found only in a single copy in all evolutionary nodes examined here and is represented by IAA33 in *A. thaliana*. In the ancestor of euphyllophytes, gene duplication events gave rise to three Aux/IAAs, which were retained in the ancestral seed plants (*Figure 4* and *Figure 4—figure supplement 1*). Common ancestor of angiosperms have 11 Aux/IAA proteins, which is more than triple the number found in gymnosperms (*Figure 4* and *Figure 4—figure supplement 1*). Finally, in addition to the ancient ncIAA generated in a first duplication event, several independent later events generated non-canonical family members lacking domains. For example, the lack of domain II in IAA20, IAA30, IAA31, IAA32, and IAA34 of *A. thaliana* appears to be an independent loss in their respective lineages in the core angiosperms (*Figure 4—figure supplement 1*).

Our data indicated that ancestral charophyte green algae had one common ancestor for both auxin (TIR1/AFB) and JA (COI1) F-box co-receptors, and following duplication in the ancestor of all land plants, developed into two independent receptors (*Figure 4* and *Figure 4—figure supplement 2*). The common ancestor of bryophytes and lycophytes had a single orthologue of *A. thaliana* TIR1/AFBs. Gene duplication events in the ancestor of euphyllophytes gave rise to three subgroups; one leading to TIR1/AFB1-3, one leading to AFB4/5 and another which is widely present in many species including the angiosperms, but has been lost in some monocots and dicots including *A. thaliana* (*Figure 4* and *Figure 4—figure supplement 2*).

Thus, our analysis of the patterns of diversification in the ARF, Aux/IAA and TIR1/AFB families identifies the auxin response complement at each evolutionary node, and in addition reveals deeply conserved non-canonical family members. Notably, many changes occurred in the composition of NAP from the common ancestor of lycophytes to euphyllophytes, which may have led to complex auxin response.

## Multi-species comparative transcriptome analysis reveals evolution of response complexity

The complements of auxin response components identified from phylogenomic analysis allow for clear predictions of which species possess a functional transcriptional auxin response system. Based on our predictions, only land plants should be able to respond. In addition, it is intuitive that the number of components in auxin response will relate to the complexity of response, but as yet there is no experimental basis for such relationship. To experimentally address the competence of species to respond to auxin, and to explore the relationship between auxin response components and the qualitative and quantitative aspects of auxin response, we performed comparative transcriptome analysis. We selected six species that belong to different ancient lineages and that each have a different complement of auxin response components (*Figure 5A*). We used the charophyte algae *Klebsormidium nitens* and *Spirogyra pratensis*, the hornwort *Anthoceros agrestis*, the liverwort *Marchantia polymorpha*, the moss *Physcomitrella patens,* and the fern *Ceratopteris richardii*. To detect only early transcriptional responses, we treated plants with auxin for 1 hr, and performed RNA-seq followed by de novo transcriptome assembly and differential gene expression analysis. To avoid inactivation of the natural auxin IAA by conjugation or transport, we treated with 10 μM of the synthetic auxin 2,4-dichlorophenoxyacetic acid (2,4-D). This compound was shown to behave like IAA in the context of the NAP (*Tan et al., 2007*).

Importantly, 68–90% of the differentially expressed genes (DEG) from de novo assemblies in *K. nitens, M. polymorpha* and *P. patens* matched with genome-based differential gene expression performed in parallel (*Figure 5—source data 1*), thus validating our approach.

Transcriptome analysis after prolonged auxin treatment in *P. patens* had identified a large set of auxin-responsive genes (*Lavy et al., 2016*). Indeed, we found 105 and 1090 genes to be auxin-regulated in *M. polymorpha* and *P. patens*, respectively (*Figure 5A*). Likewise, we found 159 and 413 genes to be auxin-regulated in *A. agrestis* and *C. richardii* (*Figure 5A*). Unexpectedly, despite lacking Aux/IAA and dedicated TIR1/AFB genes, both charophyte algae species showed a strong transcriptional response to 2,4-D treatment. A total of 1094 and 1681 genes were differentially expressed in *K. nitens* and *S. pratensis,* respectively (*Figure 5A*). Thus, there is a clear transcriptional response to 1 hr of 2,4-D treatment in all species analyzed, yet the number of genes is different, with an exceptionally large number of responsive genes in charophytes. We next determined if the number of DEG correlates with gene number in each transcriptome assembly (*Figure 5—source data 2*), and found that differences in DEG among species cannot be explained by total gene number.

We next addressed whether there were differences in the characteristics of regulation. Both charophyte species showed a high percentage of gene repression. Only 37% and 33% of DEG were activated in *K. nitens* and *S. pratensis*, respectively (*Figure 5A*). In contrast, the distribution of fold change amplitude values differed between the two charophytes where *S. pratensis* showed a general shift toward larger amplitudes of regulation (*Figure 5A*). Even though the complement of auxin response proteins are different, all three bryophytes showed a similar pattern: 36–53% of DEG were activated, with very few genes showing an amplitude over 2-fold up- or down-regulation (*Figure 5A*). In contrast, 82% of DEG were activated in *C. richardii*. We also found that there was a

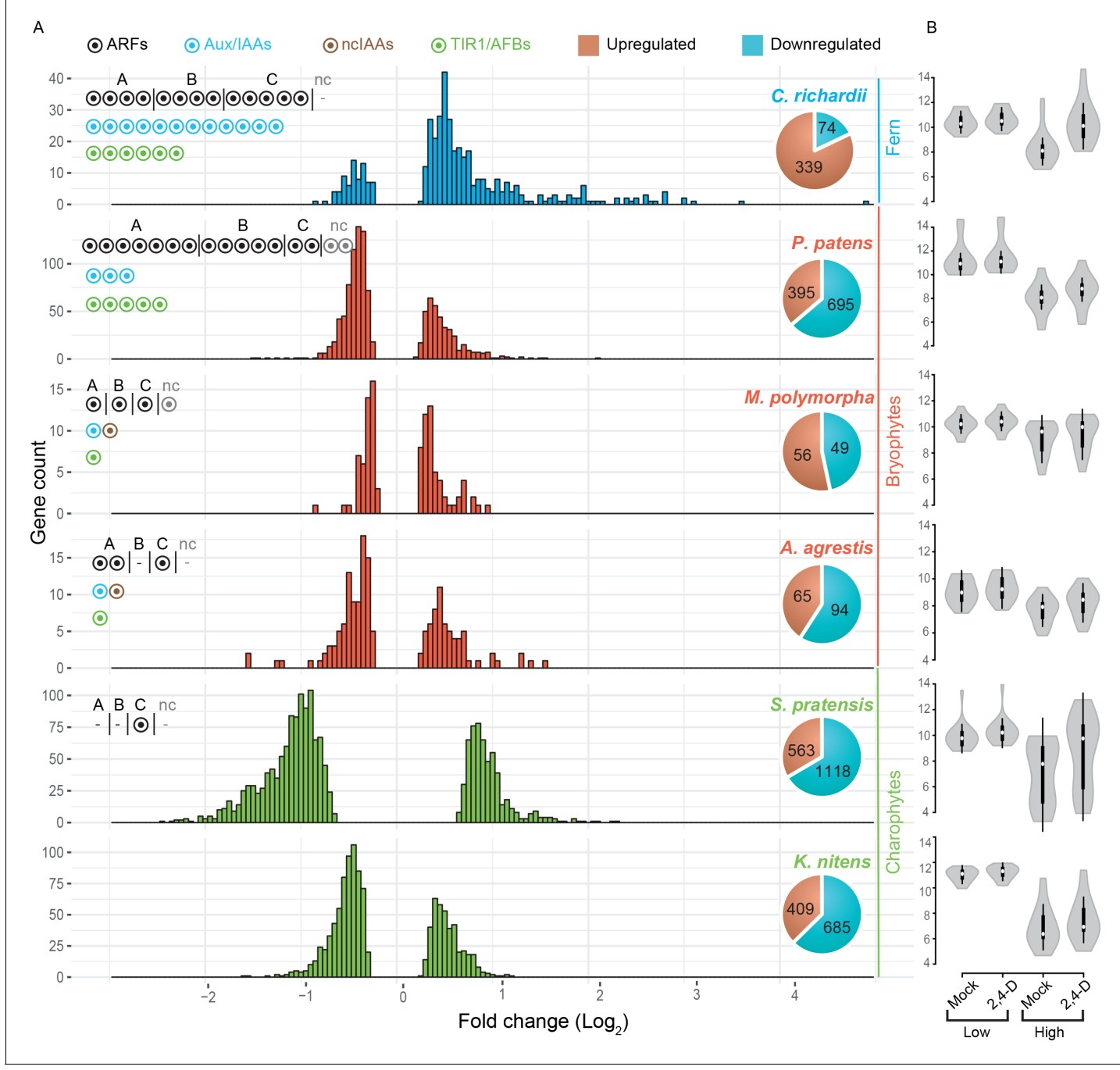

**Figure 5.** Auxin-dependent gene regulation across basal plant species. (A) Histograms represent the distribution of log$_2$ fold change among differentially expressed genes on X-axis (P$_{adj}$ <0.01). Y-axis indicates the number of genes in each log$_2$ fold-change bin. Pie charts indicate the total number of up- and down-regulated genes in each species. Circles in the top left of each graph indicate the number of NAP components. (B) Violin plots of log$_2$ normalized expression values (DEseq2-based; y-axis) of 20 least auxin activated (Low) and 20 top-most auxin upregulated (High) genes in each six species. White dot indicated the median expression value.

DOI: https://doi.org/10.7554/eLife.33399.014

The following source data is available for figure 5:

**Source data 1.** Number of DEG in de novo assembly- or genome-based transcriptome analysis.
DOI: https://doi.org/10.7554/eLife.33399.015

**Source data 2.** Summary statistics of comparative RNA-seq analysis.
DOI: https://doi.org/10.7554/eLife.33399.016

notable difference in the distribution of fold-change values, with a larger fraction of genes being more strongly activated (maximum 28 fold; *Figure 5A*).

We found that the number of auxin-responsive genes is positively correlated with the number of ARFs in land plants as seen in the expanded number of ARFs and DEG in *P. patens* and *C. richardii*. A switch to gene activation is not correlated with the number of ARFs, but rather with a duplication in the class A-ARFs in the ancestor of euphyllophytes and/or increase of Aux/IAA and TIR1/AFB. The increase in amplitude of auxin-dependent gene regulation in *C. richardii* could be a consequence of higher activation upon treatment, increased repression in the absence of auxin, or both. To determine its basis, we compared normalized expression values for the 20 top-most auxin activated, and the 20 least auxin activated genes in all species (*Figure 5B*). This revealed that the increased amplitude of the top-most activated genes in *C. richardii* is not correlated with increased expression in the presence of auxin, but rather caused by reduced expression in its absence. This quantitative property of the auxin response system is correlated with the increased numbers of Aux/IAA genes.

## Identification of a deeply conserved auxin-dependent gene set in land plants

Given that the mechanism of auxin response is ancient and conserved among all land plants, a key question is whether responses in different species involve regulation of a shared set of genes. To address this question, we performed tBLASTx searches among all DEG in our comparative transcriptome data and visualized the network of their similarities (*Figure 6—figure supplements 1* and *2*). Even though BLAST filtering is not sufficient to distinguish orthology groups in large families such as kinases, we could identify several gene families to be commonly regulated by auxin in different land plants species. Classical primary auxin-responsive genes—the Aux/IAA, GH3 and SAUR families—were shown to be auxin responsive in many angiosperm species (*Abel and Theologis, 1996*). We found different bryophyte species to show auxin-dependence in only some of these three gene families (*Figure 6A*), yet no species showed regulation of all three gene families. In contrast, *C. richardii* displayed auxin-dependence of members of all three gene families (*Figure 6A*). Given that the Aux/IAA and GH3 proteins themselves regulate auxin levels or response, this indicates that a robust feedback mechanism involving all these gene families did not exist prior to the emergence of vascular plants, and bryophytes might have different feedback mechanism.

In addition, we identified the members of class II homeodomain-leucine zipper (*C2HDZ*) and *WIP* families to be commonly activated by auxin in all land plants in our RNA-seq (note that no *WIP* gene was identified in the *A. agrestis* assembly). Indeed, qPCR analysis confirmed auxin-activation of *C2HDZ* (*Figure 6B*). We also identified the members of auxin biosynthesis gene *YUCCA* (*YUC*) family to be commonly down-regulated among multiple land plant species (except *A. agrestis*), and qPCR analysis demonstrated this to be true in *A. agrestis*, as well (*Figure 6B*). It is known that some members of *C2HDZ*, *WIP*, and *YUC* families in *A. thaliana* are also up- or down-regulated by auxin, respectively (*Crawford et al., 2015*; *Sawa et al., 2002*; *Takato et al., 2017*). While homologues of *C2HDZ* were detected in the charophyte assemblies, none was regulated by auxin, which supports the different nature of the auxin response system in these species. In summary, land plants share a deeply conserved set of auxin up- and down-regulated genes.

## Contributions of ancient components to auxin response

Our phylogenomic analysis identified several components that are deeply conserved, yet whose contributions to auxin response are unknown: two deeply conserved non-canonical auxin signaling components lack important domains (ncIAA and ncARF), while class C-ARFs diverged from all other ARFs in green algae prior to establishment of the NAP. To investigate the biological roles of these genes, we chose the liverwort *M. polymorpha,* the only genetically tractable model plant encoding *ncIAA, ncARF* and C-ARF genes. We first addressed ncIAA and ncARF function and performed CRISPR/Cas9-mediated mutagenesis (*Sugano et al., 2014*) to obtain two different alleles for each gene which presumably cause a loss-of-function by frame shift mutation (*nciaa-6, nciaa-10, ncarf-2, ncarf-10*; *Figure 7A*, *Figure 7—figure supplement 1A,B,E*). To investigate whether ncIAA and ncARF are involved in auxin response, we grew mutants on auxin-containing medium. Exogenously supplied auxin causes severe inhibition of thallus growth and increased formation of rhizoids in wild-type (*Figure 7B*; *Ishizaki et al., 2012*). *nciaa* mutants showed auxin response similar to wild-type,

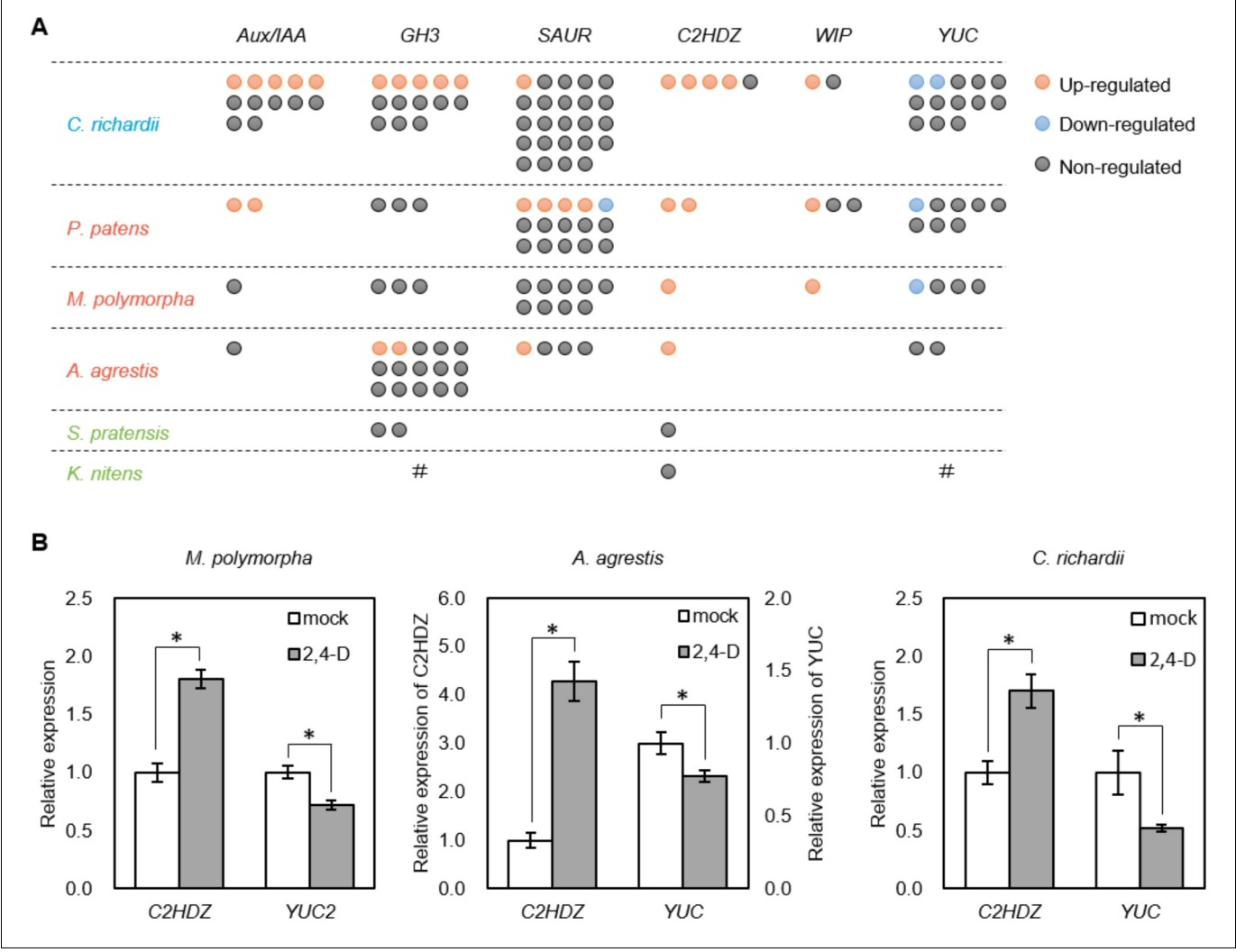

**Figure 6.** Identification of deeply conserved auxin-responsive genes. (A) Auxin-dependence of six well-known angiosperm auxin-responsive gene families (top) surveyed from de novo assembly-based transcriptomes in six species. Each circle indicates a gene copy of each gene family. Red, blue and grey circle indicate up-, down- and non-regulated genes in response to auxin. #: no homologues were identified in our transcriptome possibly due to low expression, or they might be lost during evolution. (B) qPCR analysis of conserved auxin-responsive genes. Auxin treatment was performed in the same condition with RNA-seq experiment (10 µM 2,4-D for 1 hr). Relative expression values are normalized by the expression of $EF1\alpha$ in *Marchantia polymorpha* or the amount of total RNA in *Anthoceros agrestis* and *Ceratopteris richardii*. Each bar indicates average of expression with SD (biological replicates ≥3). *: p<0.01 (t-test).

DOI: https://doi.org/10.7554/eLife.33399.017

The following figure supplements are available for figure 6:

**Figure supplement 1.** Network of up-regulated genes shared between different species upon auxin treatment.
DOI: https://doi.org/10.7554/eLife.33399.018

**Figure supplement 2.** Network of down-regulated genes shared between different species upon auxin treatment.
DOI: https://doi.org/10.7554/eLife.33399.019

while growth inhibition was strongly suppressed in *ncarf* mutants although rhizoid formation was still promoted by auxin (*Figure 7B*). We next selected two auxin-up-regulated genes (*EXP* and *WIP*) and one auxin-down-regulated gene (*YUC2*; *Eklund et al., 2015*), and examined their expression in all mutants by qPCR analysis (*Figure 7C*). In *nciaa* mutants, the expression of auxin-up-regulated genes responded similarly to the wild-type, while the expression of the auxin-repressed *YUC2* gene was significantly reduced in the absence of auxin, but similarly repressed by auxin. In *ncarf* mutants, the basal expression of auxin-upregulated genes was similar to WT, while the expression after auxin

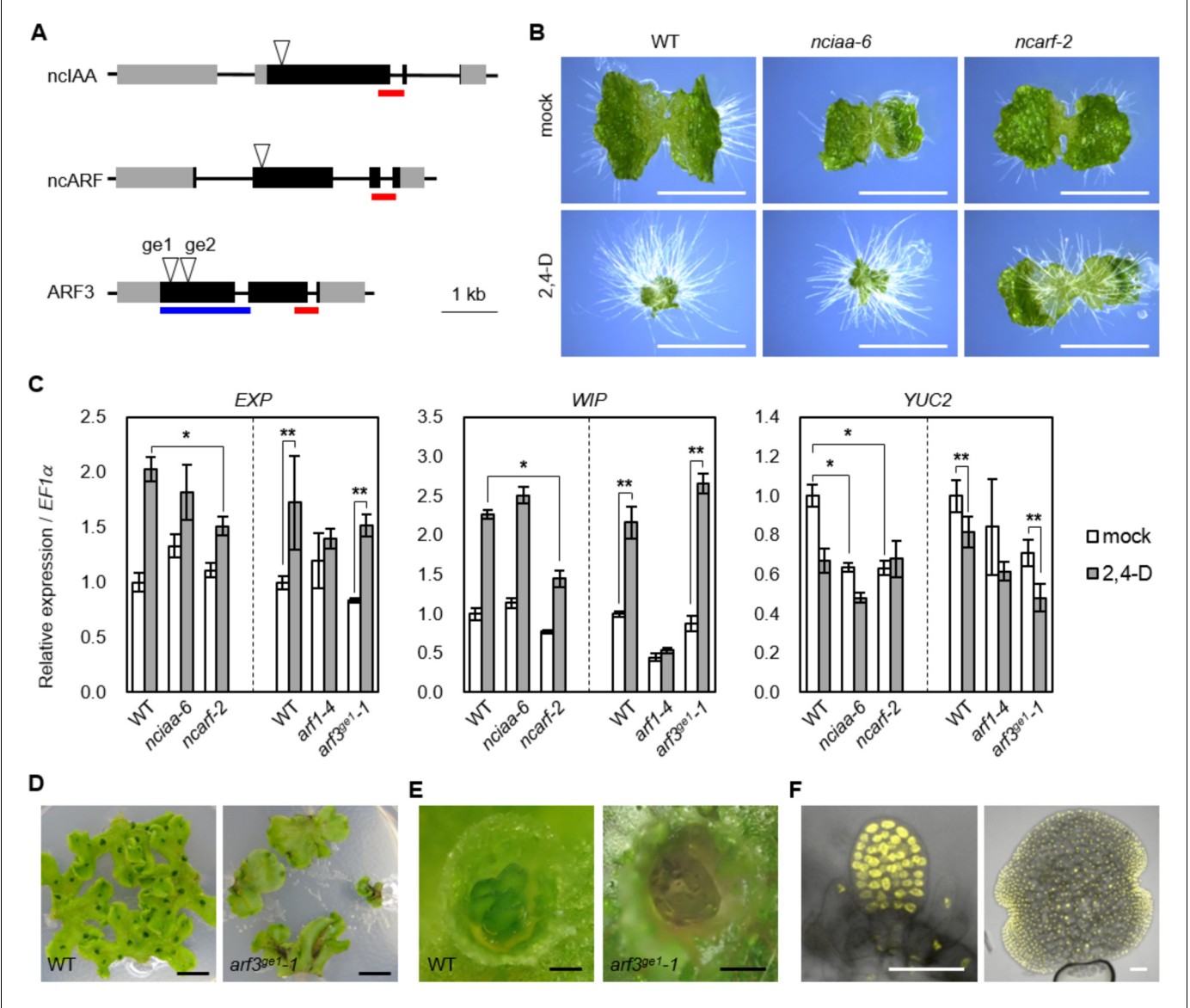

**Figure 7.** Genetic analysis of ancient components in *Marchantia polymorpha*. (**A**) Diagrams of gene structure and CRISPR/Cas9-mediated mutation in *ncIAA, ncARF* and ARF3 loci. Arrowheads indicate sgRNAs target sites. Gray and black boxes indicate UTR and CDS, respectively. Red and blue bars indicate the region coding PB1 and DBD. (**B**) 10-day-old gemmalings grown without or with 3 μM 2,4-D. Scale bars: 5 mm. (**C**) Expression analysis of auxin-responsive genes in WT, *nciaa, ncarf,* and *arf3* mutants by qPCR. 10-day-old gemmalings (*nciaa* and *ncarf*) or regenerating thalli (*arf1* and *arf3*) were treated with 10 μM 2,4-D for 1 hr. Each bar indicates average ±SD (biological replicates = 3). Asterisks indicate significant differences. *: $p < 0.01$ (Tukey test), **: $p < 0.05$ (t-test). (**D, E**) Thallus tips grown for 2 weeks (**D**) and gemma cups (**E**) of WT and *arf3^{ge1}-1* mutant. *arf3^{ge1}-1* showed growth retardation and no mature gemmae, similar to the other alleles. (**F**) Expression analysis of *_{pro}ARF3:ARF3-Citrine* in *arf3^{ge2}-1* background. Left and right panel show developing and mature gemmae, respectively. Scale bars: 5 mm in (**B and D**), 0.5 mm in (**E**), 50 μm in (**F**).

DOI: https://doi.org/10.7554/eLife.33399.020

The following figure supplement is available for figure 7:

**Figure supplement 1.** CRISPR/Cas9-mediated mutagenesis in *M.*

DOI: https://doi.org/10.7554/eLife.33399.021

treatment was significantly reduced in the mutants. The expression of *YUC2* was reduced in mock condition and auxin treatment did not change the expression. Thus, in *M. polymorpha*, ncIAA may have a function in gene expression, but is not critical for auxin response itself. On the other hand,

ncARF represents a novel positive regulator of both auxin-dependent gene activation and repression.

Finally, we focused on C-ARF function. While partial mutants have been reported in *A. thaliana*, no plants completely lacking C-ARF have been described. We used CRISPR/Cas9 gene editing to generate a series of loss-of-function mutants in MpARF3, the single C-ARF of *M. polymorpha* (*arf3$^{ge1}$-1, arf3$^{ge1}$-2, arf3$^{ge2}$-1*; *Figure 7A* and *Figure 7—figure supplement 1C,D*). All three *arf3* mutants showed dramatic defects in development, notably in vegetative propagules (gemmae) which arrested before maturation, consistent with ubiquitous ARF3 protein accumulation in these structures (*Figure 7D–F*, and *Figure 7—figure supplement 1G*). A previous study reported characterization of mutants in the class A-ARF in *M. polymorpha* (*arf1-4*) and showed that ARF1 is important for auxin response (*Kato et al., 2017*). *arf1-4* produces narrower and twisted thallus which is distinct from flat thallus of *arf3* mutants. In addition, previous studies also showed that gemma development was regulated by Aux/IAA and the class A-ARF (*Kato et al., 2015*; *Kato et al., 2017*), and we hence tested if transcriptional responses to auxin were altered in *arf3* mutants. Strikingly, all auxin-responsive genes we tested showed similar responses in WT and *arf3* mutants, while *arf1* mutants showed no auxin responses (*Figure 7C*). This result suggests that, class C-ARF in *M. polymorpha* have different target genes from A-ARF and may not be critical for auxin-dependent gene regulation.

## Discussion

### Deep origin of nuclear auxin response in the ancestor of land plants

Phylogenetic analysis and domain structural analysis provided many insights into the origin of NAP and its evolutionary trajectory. All subdomains of dedicated auxin-response proteins were recovered in transcriptomes from red algae and chlorophytes, but the multidomain protein appears only in the charophyte and land plant lineage. These findings show that proto-ARF transcription factor was established during the evolution of ancestral charophytes by combining existing domains. However, given that no defined Aux/IAA and TIR1/AFB could be identified in charophytes, a complete nuclear auxin response system is limited to land plants. Ancestors of TIR1/AFB and COI1 co-receptors could be identified in charophytes, but detailed residue analysis suggested these to be neither auxin- nor JA receptor. Thus, duplication of this gene, as well as multiple mutations in the LRR domain, must have preceded the deployment of these proteins as co-receptors. Auxin-dependence of ARFs is mediated by auxin-triggered degradation of Aux/IAA proteins, bridging ARF and TIR1/AFB proteins through two protein domains: the ARF-interacting PB1 domain and the TIR1/AFB-interacting domain II. We did find charophyte PB1-containing proteins that form a sister clade of land plant Aux/IAA. However, domain II was not detected in these proteins. Along with innovations in the proto-TIR1/AFB/COI1 protein, gain of a minimal degron motif in the Aux/IAA precursor likely completed the auxin response system in the early ancestor of land plants. Whether proto-TIR1/AFB/COI1 interact with Aux/IAA-like protein via an unknown ligand would be an interesting question for future analysis.

### Auxin responses in algal species

Despite the lack of defined Aux/IAA and TIR1/AFB auxin co-receptor, the charophytes *K. nitens* and *S. pratensis* showed an extensive transcriptional response to exogenously supplied 2,4-D within 1 hr. A recent independent study showed IAA-dependent gene expression in *K. nitens* upon prolonged treatment with higher concentrations (100 µM for 10 hr to 7 days; *Ohtaka et al., 2017*). While *S. pratensis* has a proto-C-ARF, *K. nitens* does not appear to have proto-ARFs. Thus, by definition this response system must be different from the land plant auxin response system. Indeed, the charophyte orthologue of core land plant auxin responsive genes (C2HDZ) did not respond to 2,4-D and IAA. There was little, if any, overlap between auxin-responsive transcripts in the two charophytes, and in qPCR experiments on individual genes we noticed a high variability between experiments (not shown). Thus, it appears that charophytes do respond to auxin-like molecules, but this response may not be robust, or it may strongly depend on growth conditions. Auxin resembles indole and tryptophan, and it is possible that the response to auxin observed is in fact a metabolic response to nutrient availability. Presence of endogenous IAA is observed in a wide range of algal species including charophytes, chlorophytes, rhodophytes, chromista, and cyanobacteria (*Žižková et al., 2017*).

Moreover, non-photosynthetic bacteria and fungi produce IAA and use it for communication with plants and algae (*Amin et al., 2015*; *Fu et al., 2015*), and thus it is likely that a response mechanism independent of the NAP exists in these species.

## Function of the ancestral ARFs

Our data clearly indicate that ARF transcription factors were established in common ancestor of charophyte green algae and land plants. Structural homology models suggest that all the important residues for DNA-binding are conserved in proto-ARFs, suggesting that these should bind the same target DNA sequences. This should be assessed by biochemical experiments in the future. Given that there is a core set of auxin-regulated genes shared in all land plants, an intriguing possibility is that proto-ARFs already regulated this core set of genes that only became auxin-dependent upon establishment of TIR1/AFB and Aux/IAA proteins. Identification of the transcriptional targets of these proto-ARFs should help address this question. In any event, proto-ARFs—as well as critical residues for DNA binding—have been retained in many algal genomes for hundreds of millions of years, which suggests that they perform a biologically relevant function. Whether this function is related to the processes that auxin controls in land plants is an open question.

Interestingly, our phylogenetic analysis indicated that the split between class C- and class A/B-ARFs occurred in charophytes before the establishment of Aux/IAA-TIR1/AFB co-receptor, and by extension likely before proto-ARFs were auxin-dependent. This suggests that class C-ARFs are fundamentally different from class A/B-ARFs. Indeed, genetic analysis in *M. polymorpha* revealed that its C-ARF likely does not act in auxin-dependent gene regulation. Several studies in *A. thaliana* showed that C-ARFs are involved in auxin response but the proposed role was different between studies (*Ding and Friml, 2010*; *Liu et al., 2010*; *Mallory et al., 2005*; *Wang et al., 2005*; *Yang et al., 2013*). In addition, C-ARFs of *A. thaliana* generally have weak affinity to Aux/IAA proteins (*Piya et al., 2014*). To clarify the function of this ancient ARF subfamily, auxin-responsiveness of C-ARF proteins and relationship with A- or B-ARFs should be investigated in different species.

## Novel components in auxin response

A surprising outcome of the phylogenomic analysis was the discovery of two deeply conserved non-canonical proteins: ncIAA and ncARF. Charophytes have an Aux/IAA-like protein containing a PB1 domain, but lacking domain II, which is critical for auxin perception. This protein could regulate the function of proto-ARF (or proto-RAV), but not in an auxin-dependent manner. While the canonical Aux/IAA gave rise to a large gene family, the ncIAA clade represented by a single member in every evolutionary node. The retention of a single ncIAA gene across plants suggests a fundamental function. Unfortunately, our mutant analysis in *M. polymorpha* could not reveal the function of ncIAA in auxin response and development in vegetative phase. ncIAA might have a function only in other developmental stages, or under specific stress conditions or environmental signals. No mutant in the *Arabidopsis* IAA33 gene has yet been reported, and perhaps such a mutant will help understand the ancient function of this protein.

This work revealed that a class A-ARF-derived ncARF subfamily lacking a DBD is evolutionarily conserved among liverworts, mosses, and lycophytes. Mutant analysis using *M. polymorpha* clearly showed that ncARF functions as positive regulator in transcriptional auxin responses. There are two hypothetical models for ncARF function. (1) ncARF protects canonical ARFs from AUX/IAA-mediated inactivation through the interaction of PB1 domain. (2) ncARF interacts with target gene loci by interaction with canonical ARFs and help activate expression by recruiting co-factors. Irrespective of the mechanism of ncIAA and ncARF function, future models of auxin response will need to incorporate these conserved components.

## Functional impact of increased complexity in NAP components

Through comparative transcriptomics, we infer that the number of DNA-binding ARF transcription factors scales with the number of auxin-regulated genes. Both *P. patens* and *C. richardii* have an expanded set of ARFs and display substantially more auxin-responsive genes than *A. agrestis* and *M. polymorpha*. It is likely that later duplications in the ARF family in the seed plants led to the thousands of auxin-responsive genes in these species (*Paponov et al., 2008*).

Another key evolutionary change is the transition from mostly gene repression to gene activation. We infer that this transition occurred in a common ancestor of euphyllophytes, and transcriptome analysis in *A. thaliana* and *O. sativa* shows this pattern persists in angiosperms (*Jain and Khurana, 2009*; *Paponov et al., 2008*). There is a defining difference between bryophyte and euphyllophyte ARF families—a persisting duplication in the class A-ARFs. We hypothesize that the euphyllophytes duplication created an ARF copy that is more potent, or perhaps even specialized for gene activation. However, we cannot exclude the possibility that the difference in endogenous auxin levels or tissue complexity among species may results in different sensitivity to auxin treatment.

The comparative transcriptomics also adds an interesting twist to our understanding of the functional distinction among ARF classes. Class A-ARFs are considered activators, and class B-ARFs repressors, perhaps through competing with class A-ARFs (*Lavy et al., 2016*; *Ulmasov et al., 1999*). Despite a complete lack of class B-ARFs, the hornwort *A. agrestis* showed comparable auxin-dependent gene repression to the other bryophytes, suggesting that auxin-dependent gene repression may not be mediated by class B-ARFs. Based on these findings, the role of class B-ARFs in auxin response may need to be reconsidered.

A remarkable difference between bryophyte and euphyllophyte auxin-dependent transcriptomes is the appearance of genes with a large amplitude of regulation in the latter. Many auxin-responsive genes that were first identified in angiosperms such as *A. thaliana* have very high amplitudes (*Lee et al., 2009*), but this appears to be a later innovation in the response system. The high amplitude is caused by more effective repression of gene activity in the no-auxin state, a property that is likely mediated by Aux/IAA proteins. Indeed, ferns have a much larger set of Aux/IAA proteins, as do all seed plants, and we propose that expansion of the Aux/IAA family enabled plants to articulate a clear distinction between on and off states in auxin response. In summary, this analysis reveals several design principles of the auxin response system.

## Materials and methods

**Key resources table**

| Reagent type (species) or resource | Designation | Source or reference | Identifiers | Additional information |
| --- | --- | --- | --- | --- |
| Strain, strain background (*Anthoceros agresitis*) | Oxford | PMID: 26146510 | | |
| Strain, strain background (*Ceratopteris richardii*) | Hnn | PMID: 25886741 | | |
| Strain, strain background (*Klebsormidium nitens*) | NIES-2285 | National Institute of Environmental Studies (Japan) | | |
| Strain, strain background (*Physcomitrella patens*) | Grandsden | PMID: 18079367 | | |
| Strain, strain background (*Spirogyra pratensis*) | UTEX928 | The University of Texas at Austin | | |
| Strain, strain background (*Marchantia polymorpha*) | Tak-1 | PMID: 26020919 | | |
| Genetic reagent (*M. polymorpha*) | *ncarf-2* | this paper | | mutant of ncARF of *M. polymorpha*, Tak-1 background |
| Genetic reagent (*M. polymorpha*) | *ncarf-10* | this paper | | mutant of ncARF of *M. polymorpha*, Tak-1 background |
| Genetic reagent (*M. polymorpha*) | *nciaa-6* | this paper | | mutant of ncIAA of *M. polymorpha*, Tak-1 background |

*Continued on next page*

*Continued*

| Reagent type (species) or resource | Designation | Source or reference | Identifiers | Additional information |
|---|---|---|---|---|
| Genetic reagent (*M. polymorpha*) | *nciaa-10* | this paper | | mutant of ncIAA of *M. polymorpha*, Tak-1 background |
| Genetic reagent (*M. polymorpha*) | *arf3<sup>ge1</sup>-1* | this paper | | mutant of ARF3 of *M. polymorpha*, Tak-1 background |
| Genetic reagent (*M. polymorpha*) | *arf3<sup>ge1</sup>-2* | this paper | | mutant of ARF3 of *M. polymorpha*, Tak-1 background |
| Genetic reagent (*M. polymorpha*) | *arf3<sup>ge2</sup>-1* | this paper | | mutant of ARF3 of *M. polymorpha*, Tak-1 background |
| Genetic reagent (*M. polymorpha*) | *arf1-4* | PMID: 29016901 | | |
| Recombinant DNA reagent | pMpGE_En03 | Addgene | 71535 | |
| Recombinant DNA reagent | pMpGE_010 | Addgene | 71536 | |
| Recombinant DNA reagent | pHKDW081 | this paper | | entry vector containing sgRNA for *nciaa* |
| Recombinant DNA reagent | pHKDW084 | this paper | | entry vector containing sgRNA for *ncarf* |
| Recombinant DNA reagent | pHKDW004 | this paper | | entry vector containing sgRNA for *arf3<sup>ge1</sup>* |
| Recombinant DNA reagent | pHKDW005 | this paper | | entry vector containing sgRNA for *arf3<sup>ge2</sup>* |
| Commercial assay or kit | TRIzol reagent | Thermo Fisher | 15596018 | |
| Commercial assay or kit | RNeasy Plant Mini kit | QIAGEN | 74904 | |
| Commercial assay or kit | RNase-free DNase I set | QIAGEN | 79254 | |
| Commercial assay or kit | iScript cDNA Synthesis Kit | Bio-Rad | 1708891 | |
| Commercial assay or kit | iQ SYBR Green Supermix | Bio-Rad | 1708886 | |
| Chemical compound, drug | Gamborg B5 medium | Duchefa Biochemie | G0209 | |
| Software, algorithm | BLAST + v2.2.28 | PMID: 20003500 | | |
| Software, algorithm | TransDecoder (ver2.0.1) | | | |
| Software, algorithm | InterProScan database (ver5.19–58.0) | PMID: 24451626 | | |
| Software, algorithm | MAFFT | PMID: 23329690 | | |
| Software, algorithm | Phyutility (ver2.2.6) | PMID: 18227120 | | |
| Software, algorithm | PartitionFinder (ver1.1.1) | PMID: 22319168 | | |
| Software, algorithm | RAxML (ver8.1.20) | PMID: 24451623 | | |
| Software, algorithm | iTOL (ver3) | PMID: 27095192 | | |
| Software, algorithm | Trinity | PMID: 23845962 | | |
| Software, algorithm | Bowtie2 | PMID: 22388286 | | |
| Software, algorithm | Corset | PMID: 25063469 | | |
| Software, algorithm | DEseq2 | PMID: 25516281 | | |

*Continued on next page*

*Continued*

| Reagent type (species) or resource | Designation | Source or reference | Identifiers | Additional information |
|---|---|---|---|---|
| Software, algorithm | ClustalOmega | PMID: 21988835 | | |
| Software, algorithm | Esprit | PMID: 24753421 | | |
| Software, algorithm | Modeller v9.17 | PMID: 27322406 | | |
| Software, algorithm | PyMOL | Schrödinger | | |
| Software, algorithm | Cytoscape | PMID: 14597658 | | |

## Plant materials and culture condition

Male *M. polymorpha* strain Takaragaike-1 (Tak-1) was used as wild type and cultured as described previously (*Kato et al., 2015*). *K. nitens* (NIES-2285), *P. patens* (Gransden), and *A. agrestis* (Oxford; *Szövényi et al., 2015*) were cultured on BCD medium (*Cove et al., 2009*) solidified with 1% agar under the same condition with *M. polymorpha. S. pratensis* (UTEX928) was cultured on Guillard's Woods Hole medium (*Nichols, 1973*), pH7.9 containing 1% agar under white light with a 16 hr light/8 hr dark cycle at 22°C. *C. richardii* (Hn-n) was cultured on C-fern medium (*Plackett et al., 2015*) under continuous white light at 28°C.

## Data used

Data access to 1000 plant transcriptomes was provided by the OneKP consortium (www.onekp.com; *Matasci et al., 2014*). All the transcriptome assemblies of the species from red algae, green algae, bryophytes, lycophyes, monilophytes, gymnosperms and basal angiosperms that were safely identified as non-contaminated has been used for this analysis (*Supplementary file 1*). CDS and protein sequences encoding all the orthologous genes in the three (ARF, Aux/IAA amd TIR1/AFB) gene families from *M. polymorpha*, *P. patens*, *Amborella trichopoda*, *Oryza sativa*, *Zea mays*, *Solanum lycopersicum* and *A. thaliana* were obtained from Phytozome ver11 (phytozome.jgi.doe.gov/pz/portal.html). Aux/IAA genes from *Picea abies* were obtained from Spruce Genome Project (www.congenie.org). *K. nitens* genome information was accessed from *Klebsormidium nitens* NIES-2285 genome project (*Hori et al., 2014*).

## Phylogeny construction

BLAST database for all the selected species were generated using '*makeblastdb*' module in BLAST +v2.2.28 (https://blast.ncbi.nlm.nih.gov). Protein sequences from *A. thaliana*, *M. polymorpha* and *P. patens* were used to query each database independently for each gene family using tBLASTn. All the scaffolds with the BLAST hits were extracted from the respective transcriptomes and further translated using TransDecoder (ver2.0.1; http://transdecoder.github.io). This provided the CDS and protein sequences of all the scaffolds of the BLAST hits to any of the query sequences. The protein sequences were run through the InterProScan database (ver5.19–58.0; http://www.ebi.ac.uk/interpro/) to look for conserved domains. Filtered sequences were further tested by BLASTx search against *A. thaliana* proteome to confirm orthology inferences. Some PB1 domain sequences in Chlorophytes that showed low similarity to *A. thaliana* proteins were also compared with *M. polymorpha* sequences to ascertain orthology. MAFFT (ver7.123b; *Katoh and Standley, 2013*) iterative refinement algorithm (E-INS-i) was used to align the CDS sequences. Alignment positions with more than 50% gaps were removed using the Phyutility program (ver2.2.6; http://blackrim.org/programs/phyutility/) before the phylogeny construction. PartitionFinder (ver1.1.1; *Lanfear et al., 2012*) was used to identify the most suitable evolutionary model for all the three gene families using the complete trimmed alignments on all the domains. Maximum likelihood algorithm implemented in RAxML (ver8.1.20; *Stamatakis, 2014*) with General Time Reversible (GTR) model of evolution under GAMMA rate distribution with bootstopping criterion (up to a maximum of 1000 bootstraps) was used for the phylogenetic analysis. Obtained trees were visualized using the iTOL (ver3; http://itol.embl.de/) phylogeny visualization program. Phylogenetic trees were cleaned up manually for misplaced sequences as well as for clades with long branch attraction.

## Auxin treatment

*M. polymorpha* gemmae or thallus explant without meristem and *A. agrestis* small thalli were planted on the medium covered with nylon mesh (100 µm pore) and grown for 10 days. *P. patens* protonematal tissues were grown on the medium covered with cellophane for 10 days. Sterilized spores of *C. richardii* were grown for 2 weeks after which fertilization was performed by adding 5 ml of water on the plate. Seven days after fertilization, prothalli carrying sporophytic leaves were transferred on the medium covered with nylon mesh and grown for a further 7 days, after which sporophytes contained 3–4 leaves. After growing, plants with mesh or cellophane were submerged into liquid medium and cultured for 1 day. After pre-cultivation, 2,4-D was added to a final concentration of 10 µM and plants were incubated for 1 hr. Excess liquid medium were removed with paper towels and plants were frozen in liquid nitrogen. *K. nitens* and *S. pratensis* were streaked on solid medium and grown for 2 weeks. Algal cells were collected into 40 ml of liquid medium and cultured for 1 day with shaking at ~120 rpm. Then 2,4-D was added so that final concentration became 10 µM, followed by incubation for 1 hr with shaking. After auxin treatment, algal cells were collected using filter paper and frozen in liquid nitrogen.

## RNA extraction and sequencing

Frozen plant sample were grinded into fine powder with mortar and pestle. RNA from *K. nitens, S. pratensis*, *M. polymorpha*, and *P. patens* were extracted using Trizol Reagent (Thermo Fisher Scientific; Waltham, Massachusetts) and RNeasy Plant Mini Kit (QIAGEN; Venlo, the Netherlands). RNA from *A. agrestis* and *C. richardii* were extracted using Spectrum Plant Total RNA Kit (Sigma-Aldrich). Total RNA was treated with RNase-free DNase I set (QIAGEN) and purified with RNeasy MinElute Clean Up Kit (QIAGEN). RNA-seq library construction with TruSeq kit (Illumina; San Diego, California) and 100 bp paired-end sequencing with Hiseq4000 (Illumina) were performed by BGI Tech Solutions (Hong Kong).

## Quantitative RT-PCR

cDNA was synthesized with iScript cDNA Synthesis Kit (Bio-Rad; Hercules, California). Quantitative PCR was performed using iQ SYBR Green Supermix (Bio-Rad) and CFX384 Touch Real-Time PCR Detection System. A two-step cycle consisting of denaturation at 95°C for 10 s followed by hybridization/elongation at 60°C for 30 s, was repeated 40 times and then followed by a dissociation step. Three technical and biological replicates were performed for each condition. PCR efficiencies were calculated using CFX Manager (Bio-Rad) software in accordance with the manufacturer's instructions. For *Marchantia polymorpha*, relative expression values were normalized by the expression of *EF1α* (*Saint-Marcoux et al., 2015*). All primers used for the analysis are listed in *Supplementary file 3*.

## RNA-seq data analysis

Obtained raw fastq reads were checked for quality control using FastQC (www.bioinformatics.babraham.ac.uk/projects/fastqc). De novo transcriptome assemblies for all six species were generated using Trinity (http://trinityrnaseq.github.io) with default settings. To avoid any possible contamination from sequencing method and to improve the data quality, raw reads from land plants were mapped against charophyte de novo assemblies using Bowtie2 (http://bowtie-bio.sourceforge.net/bowtie2/index.shtml) in default settings and all the perfectly mapped pairs were removed, after which new assemblies were generated from pure raw read data for each species. In a similar way, contamination was removed in charophytes by mapping them against land plant transcriptome assemblies. Once the pure de novo transcriptome assemblies were generated, again Bowtie2 was used to map individual sample to the respective transcriptome assemblies using default parameters. Further, to improve the read count estimation and reduce the redundancy in Trinity transcripts, Corset (*Davidson and Oshlack, 2014*) was implemented to estimate raw read counts using the Bowtie2 mapped alignment data. The obtained raw read counts were normalized and differentially expressed genes ($P_{adj}$ <0.01) were identified using DEseq2 (*Love et al., 2014*) implemented in R Bioconductor package. All the RNAseq raw reads were deposited in NCBI Short Read Archive (SRA) under the BioProjectID: PRJNA397394 (www.ncbi.nlm.nih.gov/bioproject/397394).

## Alignments and homology modelling

All other protein alignments mentioned in the manuscript were generated using ClustalOmega (http://www.ebi.ac.uk/Tools/msa/clustalo/). Visualization of the alignments were generated using Espript (espript.ibcp.fr). Homology models were generated using Modeller v9.17 (https://salilab.org/modeller/). Modeled 3D structures were visualized using PyMol v1.7.4 (The PyMOL Molecular Graphics System, Schrödinger, LLC).

## Core auxin-responsive gene set

All the up-regulated genes' nucleotide sequences from the six species were aligned against the same sequences using tBLASTx to find the similar (orthologous) genes among various species. From these results, the BLAST hits with E-value less than 0.001 with a length of at least 30 amino acids were considered for further analysis. Moreover, these sequences were also searched for orthologues in *A. thaliana* proteome using BLASTx. Both the similarities among the six species and the orthologous gene information from *A. thaliana* were loaded into Cytoscape (www.cytoscape.org) to visualize the network of similar gene families. A similar procedure was performed for finding the commonly downregulated gene families.

## Mutant generation for *M. polymorpha*

To generate the entry clones carrying sgRNA cassette, pairs of oligo DNAs (HK001/HK002 or HK003/HK004 for ARF3, HK162/HK163 for ncARF, HK168/HK169 for ncIAA) were annealed and cloned into pMpGE_En03 (Addgene; Cambridge, Massachusetts) using *Bsa*I site. The sequence of oligo DNAs are listed in *Supplementary file 3*. Resultant sgRNA cassette were transferred into pMpGE_010 (Addgene) by LR reaction using Gateway LR Clonase II Enzyme Mix (Thermo Fisher Scientific). Transformation into Tak-1 was performed as described previously (*Kubota et al., 2013*) using Agrobacterium strain GV3101:pMp90. For genotyping, genomic DNA was extracted by simplified CTAB (cetyltrimethylammonium bromide) method (http://moss.nibb.ac.jp/protocol.html). Genomic region including target site of sgRNA was amplified with PCR using the primer set HK079/HK131 (ARF3), HK172/HK173 (ncARF) and HK174/HK175 (ncIAA), and sequenced. All primers used in this study are listed in *Supplementary file 3*.

## Expression analysis of MpARF3 protein

MpARF3 promoter fragment including 5' UTR and 3 kb up stream region was amplified with PCR using the primer set HK111/HK026 and cloned into pMpGWB307 (*Ishizaki et al., 2015*) using *Xba*I site (pJL002). Genomic CDS of MpARF3 without stop codon was amplified with PCR using the primer set HK027/028 and subcloned into pENTR/D-TOPO vector (Thermo Fisher Scientific). Mutation which confers resistant to sgRNA was introduced by PCR using primer set HK137/138. Then mutated CDS fragment was transferred into pJL002 by by LR reaction and fused with promoter and Citrine tag (pHKDW103). All primers used in this study are listed in *Supplementary file 3*. Resultant vector was transformed into *arf3^{ge2}-1* mutant thallus as described previously. Citrine signal and bright field images were captured using a Leica SP5-II confocal laser scanning microscope system, with excitation at 514 nm and detection at 520–600 nm.

## Data availability

Raw read data of RNA-seq can be accessed in NCBI Short Read Archive (ID: PRJNA397394).

## Acknowledgements

We thank Jane A Langdale for distributing plant materials of *A. agrestis* and *C. richardii,* and Jasper Lamers and Lisa Olijslager for contributing to MpARF3 analysis. We are grateful to all contributors of the OneKP project for generating a comprehensive transcriptome database, and Eric Carpenter for providing access. We thank Kuan-Ju Lu and Nicole van 't Wout Hofland for helpful comments on the manuscript. This study was supported by an EMBO Long-Term Postdoctoral Fellowship (ALTF 415–2016) to HK and a VICI grant from the Netherlands Organization for Scientific Research (NWO; 865.14.001) to DW.

## Additional information

### Funding

| Funder | Grant reference number | Author |
|---|---|---|
| Netherlands Organisation for Scientific Research | VICI 865.14.001 | Sumanth K Mutte<br>Dolf Weijers |
| European Molecular Biology Organization | ALTF 415-2016 | Hirotaka Kato |

The funders had no role in study design, data collection and interpretation, or the decision to submit the work for publication.

### Author contributions

Sumanth K Mutte, Conceptualization, Resources, Data curation, Software, Formal analysis, Validation, Investigation, Visualization, Writing—original draft, Writing—review and editing; Hirotaka Kato, Conceptualization, Resources, Data curation, Formal analysis, Funding acquisition, Validation, Investigation, Visualization, Methodology, Writing—original draft, Writing—review and editing; Carl Rothfels, Michael Melkonian, Gane Ka-Shu Wong, Resources, Writing—original draft; Dolf Weijers, Conceptualization, Supervision, Funding acquisition, Writing—original draft, Writing—review and editing

### Author ORCIDs

Sumanth K Mutte (iD) http://orcid.org/0000-0003-3376-2354
Hirotaka Kato (iD) http://orcid.org/0000-0002-8521-1450
Dolf Weijers (iD) http://orcid.org/0000-0003-4378-141X

### Decision letter and Author response

Decision letter https://doi.org/10.7554/eLife.33399.034
Author response https://doi.org/10.7554/eLife.33399.035

## Additional files

### Supplementary files

• Supplementary file 1. Species used in phylogenomic analysis.
DOI: https://doi.org/10.7554/eLife.33399.022

• Supplementary file 2. Multiple sequence alignments used in the study.
DOI: https://doi.org/10.7554/eLife.33399.023

• Supplementary file 3. Primers used in this study.
DOI: https://doi.org/10.7554/eLife.33399.024

• Supplementary file 4. Detail for network of up-regulated genes.
DOI: https://doi.org/10.7554/eLife.33399.025

• Supplementary file 5. Detail for network of down-regulated genes.
DOI: https://doi.org/10.7554/eLife.33399.026

• Transparent reporting form
DOI: https://doi.org/10.7554/eLife.33399.027

### Major datasets

The following dataset was generated:

| Author(s) | Year | Dataset title | Dataset URL | Database, license, and accessibility information |
|---|---|---|---|---|
| Mutte S, Kato H, Weijers D | 2017 | Transcriptome regulation after auxin treatment in charophytes and land plants | https://www.ncbi.nlm.nih.gov/bioproject/397394 | Publicly available at the NCBI Bioproject database (accession no: PRJNA397394) |

The following previously published dataset was used:

| Author(s) | Year | Dataset title | Dataset URL | Database, license, and accessibility information |
|---|---|---|---|---|
| OneKP consortium | 2013 | Publicly Released Thousand Plants (1kP) Read Data | http://www.onekp.com/public_read_data.html | Publicly available at the NCBI Bioproject database (accession no: PRJEB4922) |

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
