## [Decision Letter]

Thank you for submitting your article "Origin and evolution of the nuclear auxin response system" for consideration by *eLife*. Your article has been reviewed by three peer reviewers, and the evaluation has been overseen by a Reviewing Editor and Christian Hardtke as the Senior Editor.

The reviewers have discussed the reviews with one another and the Reviewing Editor has drafted this decision to help you prepare a revised submission.

Summary:

In this manuscript, the authors combined a deep phylogenomics approach with comparative transcriptomics, and structural homology modeling to analyze the origin and early evolution of the auxin response system in more than 1,000 plant species. They identified auxin response proteins and their subdomains and precursors to reconstruct the birth and evolutionary trajectory of the auxin response system. This study provides important insights into understanding early steps in the origin, evolution and design principles of the auxin response system, and tackles a fundamentally important question that will be of high interest to researchers in plant biology and genome evolution.

Essential revisions:

1) The gene identification process for inclusion within the phylogenetic trees needs to be improved by conducting both forward and backward analysis to ensure that the orthology/parology inferences assumed in the manuscript are valid. If they are not, then the trees need to be greatly expanded to test groupings explicitly.

2) The current manuscript makes explicit claims about ancestral functions and other ancestral assumptions that have minimal experimental support. As such, the authors need to better explain what alternative hypothesis may fit the data and what future work is needed to differentiate these. It is not sufficient to merely add a paragraph in the discussion to this point but instead the entire manuscript needs extensive retuning as the vast majority of explicit claims have alternative explanations and the singular model proposed is not necessarily the best model from the data.

3) The authors should consider what would happen if rather than considering the JA pathway the outgroup, they think of JA and IAA F-box perception as independent evolutions from a common mechanistic F-box perception pool.

4) In Figure 1, compared with the domain structure of classic NAP components in land plants, some of the non-canonical ARF, Aux and TIR identified in algae only contain one or two (sub-)domains. It is still unclear that whether these ancestral and weird NPA genes are only present in the algae lineages, or they also could be found in land plants and belong to other gene families. The evidence that these ancient NPA-like component is alga-specific, will strongly support that these identified genes belongs to the ARF/Aux/TIR families, and also support the view that the canonical NAP proteins with multi-domain in land plants are evolved by combining existing domains together (just like gene-fusion event) during they diverged from the alga lineages.

5) The paper focuses on analyzing the evolution of three critical NPA-related genes – ARF, Aux/IAA and TIR/AFB. However, in Figure 2, the RAV is suddenly introduced for analysis, and meanwhile, there is little information about the RAV in the manuscript, which is a bit puzzling. The evolutionary relationship should be clearly clarified between RAV/Aux/ARF and tell us how to distinguish the RAV gene from the Aux/ARF, if defined by the domain, it is better to add the domain structure of RAV protein in the supplementary data.

6) It is claimed that a TIR homologous genes is identified in Charopyhte, which might be the precursor of TIR/COI1. In Figure 3, the so-called TIR1/COI1 hybrid described in Figure 4 do not have any predicted structural similarity compared with the split TIR and COI proteins in land plants. If further evidence is provided that some specific conserved amino acids, separately located at the diverged TIR and COI1 in land plant, could be simultaneously found in this ancient TIR1/COI1 protein, it will strongly enhance the concept that this ancestral gene is a TIR1/COI1 hybrid, and then it was separated into two genes TIR and COI1 during the origin of the land plant that diverged from the charophyte.

7) In Figure 5, it is really very interesting that accompanied with the evolution and diversification of the NAP components during plant evolution, more and more auxin regulated genes tend to be downregulated, and the repression pathway begin to dominate expression pattern. However, besides the differential regulation mechanism, according to this pie chart, it is unclear that whether there is some overlap in this up- or down-regulated genes from these six plant lineages, or these up- or down-regulated genes are fully different between these ancient plant species.

8) In Figure 6, according to the transcriptomic data, the YUC gene in the A. agrestis is non-regulated by auxin. However, the qRT-PCR showed that there might be a downregulated mechanism for the YUC gene in A. agrestis after treatment. Meanwhile, the difference of the YUC expression between before and after auxin treatment is so slight, not only in the A. agrestis but also in the plants Mp and Cr, so the conclusion is not so solid. To enhance it, a series of higher concentration of auxin could be applied to check whether there is enhanced down-regulation of the YUC genes after treatment. By the way, the YUC is not identified in the S. pratensis and K. nitens, which might be due to the loss of this gene in these two, specific algae species but the homologous YUC genes are really present in the green algae lineages, even in the bacteria. So, it is by no means the YUC gene originated from the land plant, the discussion here should be much more careful.

9) In Figure 7 control that the phenotype of the arf1 mutant line should be added to make the arf3 defects much more comparable. Meanwhile, the same treatment as Figure 7 also could be extended to the arf1 and arf3 mutants, showing that the MpARF3 is important for mediating auxin signaling, rather than the MpARF1.

[Editors' note: further revisions were requested prior to acceptance, as described below.]

Thank you for resubmitting your work entitled "Origin and evolution of the nuclear auxin response system" for further consideration at *eLife*. Your revised article has been favorably evaluated by Christian Hardtke (Senior editor), a Reviewing editor, and three reviewers.

The reviewers have discussed the reviews with one another and the Reviewing Editor has drafted this decision to help you prepare a revised submission.

Summary:

This revised manuscript has addressed most of the comments raised by three reviewers. However, the authors need to further address some other criticisms raised by reviewers before the manuscript could be considered for publication.

Essential revisions:

1) Reviewer 1 is still puzzled by some of the nomenclature fuzziness. The paper is about auxin but it is never really clear that this is about indole-3-acetic acid. Especially as 2,4D is more correctly a mimic of phenylacetic acid that also has auxin activities. Although the modern development field has largely limited itself to IAA studies, the original "auxins" were not IAA. It seems that a paper on evolutionary complexity in this pathway should be very careful to cover and be precise about the metabolic complexity as well.

2) Figure 2 is a very nicely drawn illustration, but it is unclear how this was derived. There are mathematical methods to combine multiple trees, but it seems that in this case that the illustration is simply an illustration. Figure 2—figure supplement 1 does not really support the explicit shape of the tree in Figure 2 as a number of the branches being claimed as really have minimal support, often less than 60.

3) In general, for all phylogenetic trees, unsupported branches need to be shown as unresolved. There are innumerable branches that have no support but are shown as resolved. For example, in the TIR1 tree, the split between TIR1, AFB1, AFB2, AFB3 and AFB4/5 is from branches with 20 or lower support. And as such, how is the diagram in Figure 4 supported? It is an interesting hypothesis, but it is not clear how this diagram arose from the underlying trees.

---

## [Author Response]

Essential revisions:1) The gene identification process for inclusion within the phylogenetic trees needs to be improved by conducting both forward and backward analysis to ensure that the orthology/parology inferences assumed in the manuscript are valid. If they are not, then the trees need to be greatly expanded to test groupings explicitly.

To confirm the orthology of basal plant sequences to queried Arabidopsis sequences, we tested all sequence we identified in all species, that were used for ARF, IAA, TIR1/AFB trees with a BLAST search against the Arabidopsis proteome and listed the best hit. All land plant sequences showed the highest similarity with corresponding family members in Arabidopsis (as given in Figure 4). Proto-ARFs and proto-RAV sequences in Charophytes showed the highest similarity with ARF’s and RAV, respectively. B3-containing sequences in Chlorophytes showed best hits with various B3-type transcription factors. Given that Chlorophyte B3 domains are basal to all B3-type transcription factors in land plants (Swaminathan et al., 2008), this is the expected result. Aux/IAA-like sequences from *Coleochaete irregularis* did not present Aux/IAA’s or ARF’s as the best BLAST hit. We believe this to be due to the large sequence distance between Coleochaete from PB1 domains in Arabidopsis. However, when we BLAST Charophyte Aux/IAA-like sequences against the *Marchantia polymorpha* genome, the best hit is ncIAA, confirming our orthology inference. TIR1/COI1 sequences in Charophytes showed the highest similarity with either of TIR1/AFBs or COI1 (See also further points below). We have now modified Figure 1—figure supplement 1 and the Materials and methods section to clarify our forward/backward BLAST strategy.

2) The current manuscript makes explicit claims about ancestral functions and other ancestral assumptions that have minimal experimental support. As such, the authors need to better explain what alternative hypothesis may fit the data and what future work is needed to differentiate these. It is not sufficient to merely add a paragraph in the discussion to this point but instead the entire manuscript needs extensive retuning as the vast majority of explicit claims have alternative explanations and the singular model proposed is not necessarily the best model from the data.

In our manuscript, we made two explicit claims about the function of ancestral proteins. (1) Proto-ARFs might have the same DNA-binding mode as ARFs in land plants, and (2) auxin-perception through Aux/IAA and TIR1/AFB was established after the split of Charophytes and land plants.

In regard to claim 1), we presented that the amino acid sequence of DBD is extremely conserved between proto-ARFs and ARF’s in land plants (Supplementary file 2), and the proto-ARF DBD’s also follow the same 3D structure including dimerization domain (Figure 3). Based on these results we consider that the current hypothesis is the most plausible. Of course, the hypothesis should be experimentally confirmed in the future studies, hence we add a sentence this effect in the Discussion section.

With respect to claim 2), we appreciate a series of constructive comments related to the origin of TIR1 and COI1. As described in detail below (Essential revision 3), we improved the phylogenetic tree and homology model for TIR1/COI1-like proteins in Charophytes. The new data strongly suggests that the TIR1/COI1-like proteins in extant Charophytes form a common sister clade to both TIR1/AFB and COI1 in land plants, and they are unlikely to bind auxin or JA. Given that Aux/IAA-like proteins of Charophytes do not have the critical domain for auxin perception, the most plausible hypothesis is that auxin perception through TIR1/AFB and Aux/IAA is limited to land plants. We consider it is unlikely that the ancestral proteins to TIR1/COI1 and Aux/IAA perceived auxin and were lost independently during Charophyte evolution. However, we cannot exclude the possibility that Charophyte TIR1/COI-like proteins interact with the Aux/IAA-like protein via an unknown ligand. We now refer to this possibility in Discussion section.

3) The authors should consider what would happen if rather than considering the JA pathway the outgroup, they think of JA and IAA F-box perception as independent evolutions from a common mechanistic F-box perception pool.

This is an excellent suggestion, and we now realise that we made this point unnecessarily complicated due to the poorly supported phylogenetic tree in our original submission.

To improve the resolution of the TIR1/COI1 phylogeny, and to confirm the evolutionary position of TIR1/COI1-like proteins in Charophytes, we generate a new phylogenetic tree for TIR1/COI1 including other, functionally unrelated but relatively similar F-box LRR proteins (MAX2, VFB, and EBF families) from land plants (new Figure 4—figure supplement 2). In addition, we elaborated the TIR1/AFB phylogenetic tree to focus on the evolution in land plants (new Figure 4—figure supplement 3). The former tree clearly showed that none of the green algal sequences is specifically grouped with either TIR1/AFB or COI1 in land plants, and thus form a sister group to both TIR1 and COI1.

Originally, we had used Klebsormidium and Spirogyra proteins to generate homology models of TIR1/COI1 orthologues. This is because we had accesses to the (draft) genomes of these species. However, the new F-box LRR tree indicated that the Spirogyra sequence actually does not belong to the TIR1/COI1 sister clade. The Klebsormidium protein was still grouped into the sister clade of TIR1/COI1, but its protein sequence is short and very different from land plant TIR1/COI1. Therefore, we generated a new homology model using the *Coleochaetae irregularis* protein, which belongs to the sister clade of TIR1/COI1. The secondary structure of the Coleochaete protein is highly similar to that of TIR1 and COI1 including the 3_10_ helix we referred to in our original manuscript (new TIR1/COI1 alignment in Supplementary file 2). However, the overall amino acid sequence is different from both TIR1 and COI1. Out of 40 residues which are specifically conserved in either TIR1/AFB or COI1, only 7 and 11 residues are identical to TIR1/AFB or COI1, respectively (black asterisks in Supplementary file 2). Notably, 11 out of 12 hormone contacting residues of TIR1/AFB and COI1 are not conserved in the Coleochaete protein (new Figure 3 and Supplementary file 2). Based on these data, the most plausible interpretation is that TIR1/AFB and COI1 in land plants evolved from a common ancestor in charophyte algae, and it is unlikely that TIR1/COI1-like protein in charophytes bind to either auxin or jasmonate. We have now modified the manuscript (Results section) and corresponding figures to elaborate on these new findings. In addition, we have modified “TIR1/COI1 hybrid” in Figure 4 into “TIR1/COI1 precursor” for accuracy.

4) In Figure 1, compared with the domain structure of classic NAP components in land plants, some of the non-canonical ARF, Aux and TIR identified in algae only contain one or two (sub-)domains. It is still unclear that whether these ancestral and weird NPA genes are only present in the algae lineages, or they also could be found in land plants and belong to other gene families. The evidence that these ancient NPA-like component is alga-specific, will strongly support that these identified genes belongs to the ARF/Aux/TIR families, and also support the view that the canonical NAP proteins with multi-domain in land plants are evolved by combining existing domains together (just like gene-fusion event) during they diverged from the alga lineages.

We interpret this comment to refer to the sequences in red algae and Chlorophytes. In this case, indeed, the proteins containing small domains of NAP components are specific to these algae. In land plants, DD and AD are only found in the ARF family and never combined with BROMO or ARID domains. To explain these points, we added a sentence “In land plants, the DD and AD are only found in ARF family.” (subsection “Origin of nuclear auxin response components”).

In addition, Chlorophyte B3 domain-containing genes are thought to be sister to both ARF and the other B3-containing genes in land plants (Swaminathan et al., 2008). As shown in Figure 2 and Figure 2—figure supplement 1, Chlorophyte PB1-containing genes form a Chlorophyte-specific clade along with proto-RAV in Charophytes.

5) The paper focuses on analyzing the evolution of three critical NPA-related genes – ARF, Aux/IAA and TIR/AFB. However, in Figure 2, the RAV is suddenly introduced for analysis, and meanwhile, there is little information about the RAV in the manuscript, which is a bit puzzling. The evolutionary relationship should be clearly clarified between RAV/Aux/ARF and tell us how to distinguish the RAV gene from the Aux/ARF, if defined by the domain, it is better to add the domain structure of RAV protein in the supplementary data.

RAV is one of the B3-containing protein families in land plants, and it has an additional AP2-type DBD in its N-terminus. The reason why we picked up the RAV family in this study is that proto-RAV genes in Charophytes contain a PB1 domain in their C-terminus similar to ARF’s, even though land plant RAV’s do not have this PB1 domain. Domain structures of two DBD’s, phylogenetic analysis, and amino acid sequences of B3 domains clearly indicate that proto-RAV’s in Charophytes are the homologue to the land plant RAV family. For better introduction of the RAV family and better explanation of our results, we modified the manuscript (subsection “Origin of nuclear auxin response components”) and added a supplementary figure regarding the B3 domain of RAV (Figure 1—figure supplement 3).

6) It is claimed that a TIR homologous genes is identified in Charopyhte, which might be the precursor of TIR/COI1. In Figure 3, the so-called TIR1/COI1 hybrid described in Figure 4 do not have any predicted structural similarity compared with the split TIR and COI proteins in land plants. If further evidence is provided that some specific conserved amino acids, separately located at the diverged TIR and COI1 in land plant, could be simultaneously found in this ancient TIR1/COI1 protein, it will strongly enhance the concept that this ancestral gene is a TIR1/COI1 hybrid, and then it was separated into two genes TIR and COI1 during the origin of the land plant that diverged from the charophyte.

We have improved the phylogenetic tree, multiple sequence alignment and homology model of TIR1/COI1 as described above (Essential revision #3). The results show that Charophyte TIR1/COI1 precursors have a similar secondary structure to TIR1 and COI1 in land plants, form a common sister clade to both TIR1/AFB and COI1 families, and amino acids in the hormone-contacting domain do not resemble either TIR1/AFB or COI1.

7) In Figure 5, it is really very interesting that accompanied with the evolution and diversification of the NAP components during plant evolution, more and more auxin regulated genes tend to be downregulated, and the repression pathway begin to dominate expression pattern. However, besides the differential regulation mechanism, according to this pie chart, it is unclear that whether there is some overlap in this up- or down-regulated genes from these six plant lineages, or these up- or down-regulated genes are fully different between these ancient plant species.

Apparently, we did not convey the message clearly. Unlike is stated here, our comparative transcriptome analysis revealed that more genes are “activated” by auxin in vascular plants than bryophytes (not downregulated). The question of how much overlap there is among species is highly relevant and was an omission in our original manuscript. We have now performed reciprocal BLAST searches among DEG’s in all 6 species and carried out network analysis based on similarities of DEG’s across species. This helps visualising commonly regulated gene families, or orthology groups, rather than individual paralogues (New Figure 6—figure supplement 1 and Figure 6—figure supplement 2). This analysis showed that there is a number of auxin-regulated gene clusters showing some similarity. BLAST filtering is not necessarily enough to identify true paralogues and it is impossible to distinguish those in large gene family such as bHLH and kinases. Thus, it is impossible to derive an exact number or percentage of gene families that overlap across the species. Nevertheless, our analysis could identify some very specific gene families including C2HDZ, WIP and YUC to be commonly regulated by auxin in land plants. This strongly supports our conclusion that land plants share a set of auxin-responsive genes through the evolution. Furthermore, this analysis also showed that Charophyte auxin-regulated genes do not fall in the land plant clusters, and in addition revealed little overlap among the two Charophytes included here. We have now added this new data and modified the manuscript accordingly. Please see Results section and Materials and methods section for details.

8) In Figure 6, according to the transcriptomic data, the YUC gene in the A. agrestis is non-regulated by auxin. However, the qRT-PCR showed that there might be a downregulated mechanism for the YUC gene in A. agrestis after treatment. Meanwhile, the difference of the YUC expression between before and after auxin treatment is so slight, not only in the A. agrestis but also in the plants Mp and Cr, so the conclusion is not so solid. To enhance it, a series of higher concentration of auxin could be applied to check whether there is enhanced down-regulation of the YUC genes after treatment. By the way, the YUC is not identified in the S. pratensis and K. nitens, which might be due to the loss of this gene in these two, specific algae species but the homologous YUC genes are really present in the green algae lineages, even in the bacteria. So, it is by no means the YUC gene originated from the land plant, the discussion here should be much more careful.

We believe that the differential expression (which is small in amplitude, but clearly significant in qPCR) might have been masked in RNA-seq analysis during normalization or statistical analysis. Based on studies in angiosperms, 10 μM of auxin would already be a very high concentration. To test explicitly whether our choice of concentration in the experiment was suboptimal, we treated *M. polymorpha* with different concentrations of 2,4-D (1, 10, 100 μM), and determined the expression of representative auxin-responsive genes. However, 100 μM showed a weaker effect than 10 μM for three out of four genes tested, probably due to toxicity (see Author response image 1).

In Figure 6, we did not list a YUC homologue in Charophyte species, because we could not find any YUC homologues in our de novo assemblies. As pointed out in the comment, this might be caused by low expression under our growth condition, and this may also be true for GH3 in *K. nitens*. We understand the homologues of YUC and GH3 are exist in the *K. nitens* genome. To avoid confusion for readers, we have added a footnote to the legend of Figure 6 to explain that the absence of these genes in our transcriptome might be caused by low expression.

9) In Figure 7 control that the phenotype of the arf1 mutant line should be added to make the arf3 defects much more comparable. Meanwhile, the same treatment as Figure 7 also could be extended to the arf1 and arf3 mutants, showing that the MpARF3 is important for mediating auxin signaling, rather than the MpARF1.

We appreciate this comment, but we do not conclude that MpARF3 is important for mediating auxin signalling. On the contrary, we conclude that it is not involved in regulating auxin-dependent genes. With regards to the role of MpARF1, Kato et al., (2017) have reported a detailed phenotypic analysis of Mp*arf1* mutants and showed that MpARF1 is critical for auxin response. Our qPCR analysis also showed MpARF1, but not MpARF3 is important for auxin-dependent gene regulation (Figure 7). For Figure 7, similar growth assays for Mp*arf1* mutants have already been done and reported in the Kato at al. (2017) paper. Unfortunately, it is impossible to add *arf3* in Figure 7 because *arf3* mutants never produce mature gemmae, and thus we cannot determine growth responses to auxin in gemmae. We have now added an explanation in subsection “Auxin responses in algal species” about the work of Kato et al., (2017).

[Editors' note: further revisions were requested prior to acceptance, as described below.]

Essential revisions:1) Reviewer 1 is still puzzled by some of the nomenclature fuzziness. The paper is about auxin but it is never really clear that this is about indole-3-acetic acid. Especially as 2,4D is more correctly a mimic of phenylacetic acid that also has auxin activities. Although the modern development field has largely limited itself to IAA studies, the original "auxins" were not IAA. It seems that a paper on evolutionary complexity in this pathway should be very careful to cover and be precise about the metabolic complexity as well.

The reviewer is entirely right. Auxins are a diverse group of chemically related compounds that are defined by their biological activities. For the nuclear auxin pathway, it has become clear that IAA, NAA and 2,4-D bind the same receptor pocket in the same way and induce the same response. Thus, 2,4-D is often used as a stable and potent analog for experiments related to gene regulation. We now open the introduction with a statement on the chemical diversity of auxin molecules, and later in the introduction explain that these all act in the nuclear auxin pathway in comparable way. We further explain the choice of 2,4-D in our transcriptomics experiment when the experiment is introduced.

2) Figure 2 is a very nicely drawn illustration, but it is unclear how this was derived. There are mathematical methods to combine multiple trees, but it seems that in this case that the illustration is simply an illustration. Figure 2—figure supplement 1 does not really support the explicit shape of the tree in Figure 2 as a number of the branches being claimed as really have minimal support, often less than 60.

The reviewer is correct in stating that this is merely an illustration, and the branches were not drawn according to their true scale. We should stress that, the tree in Figure 2—figure supplement 1 represents a phylogeny based ONLY on the PB1 domain, a short domain with less than 50 informative amino acid positions. Thus, by definition the resolution of the phylogeny will be poor and highly sensitive to permutation in bootstrap analysis. For generating the schematic in Figure 2, we additionally used the DBD phylogeny to resolve the relationships among ARF proteins. We agree that one should not get the wrong impression from this schematic illustration. We have therefore replaced this figure panel by a schematic tree that shows only the well-resolved branches. The aim of the figure was to illustrate that Aux/IAA’s and ARF’s have a separate origin that dates back to the Charophytes. There is no doubt about these inferences based on the “real” phylogenetic tree (Figure 2—figure supplement 1).

3) In general, for all phylogenetic trees, unsupported branches need to be shown as unresolved. There are innumerable branches that have no support but are shown as resolved. For example, in the TIR1 tree, the split between TIR1, AFB1, AFB2, AFB3 and AFB4/5 is from branches with 20 or lower support. And as such, how is the diagram in Figure 4 supported? It is an interesting hypothesis, but it is not clear how this diagram arose from the underlying trees.

We appreciate this comment but disagree that “innumerable branches that have no support are shown as resolved”. We believe that the reviewer may have inspected the earlier version of the TIR1 tree, which indeed had poor bootstrap values, but was replaced in the revised version of our manuscript. The grouping of TIR1/AFB1, AFB2/AFB3, AFB4/AFB5 and the relative that was lost in the core eudicots with their orthologs from other tracheophyte lineages is very well supported (bootstrap values >65). It is true that their relative position (and hence order) is supported by relatively poor bootstrap values. However, this does not impact the grouping we summarize in Figure 4. For this reason, we purposely avoided any discussion about the order of appearance of the gene copies in duplication events. For full clarity, we have provided all our trees as open accessible files on the iTOL website, such that all readers can interact with the trees. In summary, the diagram in Figure 4 is fully supported by the trees. We now mention in the figure legend that tree topology is derived only from the well-supported branches.